# Applying an isotope-enabled regional climate model over the Greenland ice sheet: effect of spatial resolution on model bias

Marcus Breil[1], Emanuel Christner[1], Alexandre Cauquoin[2,3], Martin Werner[2], Melanie Karremann[1], Gerd Schädler[1]

[1]Institute of Meteorology and Climate Research, Karlsruhe Institute of Technology, Eggenstein-Leopoldshafen, Germany
[2]Alfred Wegener Institute, Helmholtz Centre for Polar and Marine Sciences, Bremerhaven, Germany
[3]Institute of Industrial Science, University of Tokyo, Kashiwa, Japan

*Correspondence to*: Marcus Breil (marcus.breil@kit.edu)

**Abstract.** In order to investigate the impact of spatial resolution on the discrepancy between simulated $\delta^{18}O$ and observed $\delta^{18}O$ in Greenland ice cores, regional climate simulations are performed with the isotope-enabled Regional Climate Model (RCM) COSMO_iso. For this purpose, isotope-enabled General Circulation Model (GCM) simulations with ECHAM5-wiso General Circulation Model (GCM) under present-day and MPI-ESM-wiso GCM under mid-Holocene conditions are dynamically downscaled with COSMO_iso for the Arctic region. The capability of COSMO_iso to reproduce observed isotopic ratios in Greenland ice cores for these two periods is investigated by comparing the simulation results to measured $\delta^{18}O$ ratios from snow pit samples, Global Network of Isotopes in Precipitation (GNIP) stations and ice cores. To our knowledge, this is the first time that a mid-Holocene isotope-enabled RCM simulation is performed for the Arctic region.

Under present-day conditions, a dynamical downscaling of ECHAM5-wiso (1.1° x 1.1°) with COSMO_iso to a spatial resolution of 50 km improves the agreement with the measured $\delta^{18}O$ ratios for 14 of 19 observational data sets. A further increase in the spatial resolution to 7 km does not yield substantial improvements except for the coastal areas with its complex terrain. For the mid-Holocene, a fully coupled MPI-ESM-wiso time slice simulation is downscaled with COSMO_iso to a spatial resolution of 50 km. In the mid-Holocene, MPI-ESM-wiso already agrees well with observations in Greenland and a downscaling with COSMO_iso does not further improve the model-data agreement. Despite this lack of improvements in model biases, the study shows that in both periods, observed $\delta^{18}O$ values at measurement sites constitute isotope ratios which are mainly within the subgrid-scale variability of the global ECHAM5-wiso and MPI-ESM-wiso simulation results. The correct $\delta^{18}O$ ratios are consequently not resolved in the GCM simulation results and need to be extracted by a refinement with an RCM. In this context, the RCM simulations provide a spatial $\delta^{18}O$ distribution by which the effects of local uncertainties can be taken into account in the comparison between point measurements and model outputs. Thus, an isotope-enabled GCM-RCM model chain with realistically implemented fractionating processes, constitutes a useful supplement to reconstruct regional paleo-climate conditions during the mid-Holocene in Greenland. Such model chains might also be applied to reveal the full potential of GCMs in other regions and climate periods, in which large deviations relative to observed isotope ratios are simulated.

# 1 Introduction

Stable isotopes of water (HD$^{16}$O and H$_2$$^{18}$O) are fractionated during any phase transition. This fractionating process depends on temperature (Dansgaard, 1953; Craig and Gordon, 1965; Jouzel and Merlivat, 1984), so that water isotopic ratios (expressed here in the usual δ notation, δD and δ$^{18}$O with respect to the Vienna Standard Mean Ocean Water V-SMOW) reflect the atmospheric conditions under which the fractionating process took place (Dansgaard, 1964; Merlivat and Jouzel, 1979; Gat, 1996). This process is generally utilized to reconstruct paleo-climate conditions such as past temperature changes, using isotopic ratios stored in climate archives (Dansgaard et al., 1969; Masson-Delmotte et al., 2005; Jouzel, 2013).

In Arctic regions like Greenland, ice cores constitute an exceptional climate archive. Over thousands of years, accumulated snow was solidified to ice, preserving at some locations the water isotopic ratios since the last interglacial period. Climate reconstructions based on these ice cores show that the climate conditions changed considerably in Greenland during the Holocene (here defined as the period between present-day and 12 ka; Marcott et al., 2013). Between the early Holocene and the Holocene Thermal Maximum in the mid-Holocene (6 ka), a pronounced warm phase took place. Since then, temperatures steadily decreased until the late Holocene (Marcott et al., 2013; Moossen et al., 2015). In this context, the mid-Holocene is a period of particular interest, as by that time an Arctic warming had taken place due to orbital forcing variations and their related feedbacks on large-scale climate variations, which exhibits similarities to the strong recent Arctic warming. For Greenland, the mid-Holocene provides the opportunity to investigate the processes leading to this warming in more detail and to potentially obtain new insights about the future development of the Arctic region (Yoshimori and Suzuki, 2019).

While General Circulation Models (GCMs) are generally able to reproduce the direction and large-scale patterns of past climate changes (e.g. Timm and Timmermann, 2007; Smith and Gregory, 2012), they often fail to reproduce the magnitude of regional changes (Braconnot et al., 2012; Harrison et al., 2014), documented in various local climate archives. Thus, a scale gap might exist between the measured point information and the large-scale climate information generated by GCMs. The comparison of observational and GCM data can therefore be subject to considerable uncertainties (Felzer and Thompson, 2001).

Especially for structured landscapes, the spatial resolution in GCMs is often too coarse to resolve relevant local factors (Jost et al., 2005; Fischer and Jungclaus, 2011). Important properties like topography and surface conditions are consequently only represented in a generalized and imprecise form in climate simulations. In most cases, this does not adequately represent the complex characteristics of the land surface and its associated interactions with the atmosphere. For stable water isotopes, key physical processes of isotope fractionation are therefore not well resolved in coarse resolution GCMs, leading to differences between simulated and observational isotope data, especially in complex terrains (Sturm et al., 2005; Werner et al., 2011). Isotope-enabled GCMs are consequently not able to quantitively reproduce regional changes in isotope ratios (e.g. Risi et al., 2010), and the simulated isotope ratios with GCMs exhibit in many cases larger deviations relative to observed ratios than the results of corresponding Regional Climate Model (RCM) simulations. For instance, Sturm et al., (2007) were able to reduce the bias of simulated isotope ratios in precipitation through a regional downscaling of an isotope-enabled GCM run in

South America. Comparable results were achieved by Sjolte et al., (2011) for isotope-enabled RCM simulations in Greenland.

Therefore, in the present study, isotope-enabled GCM simulation results for the Arctic region are dynamically downscaled with an isotope-enabled RCM to a higher temporal and spatial resolution. By means of such regional simulations, the spatial and temporal variability of the isotopic ratios in the Arctic is potentially increased, accounting for the heterogeneity of local

conditions at the different ice core locations and the associated uncertainties. In this way, the impact of highly resolved local conditions on the spatial and temporal variability of isotopic ratios is investigated, and the impact of such small-scale variability on the discrepancy between simulated and observed paleo-climate conditions in the Arctic region is examined.

To explore this, the isotope-enabled version of the RCM COSMO-CLM (Rockel et al., 2008), COSMO_iso (Pfahl et al., 2012; Christner et al., 2018), is used. In a first step, the general suitability of COSMO_iso to be used for isotope applications

in Greenland is assessed. For this purpose, near-surface temperatures and precipitation amounts simulated with the standard COSMO version are compared with observations in the Arctic region. Subsequently, the capability of COSMO_iso to simulate realistic water isotopic ratios for Greenland is tested by downscaling a global present-day simulation with an isotope-enabled GCM for the Arctic region. The GCM and RCM results are then compared to measured water isotope ratios in precipitation and snow pit samples. Afterwards, the tested isotope-enabled COSMO_iso model system is used to

downscale an isotope-enabled GCM simulation for a mid-Holocene time-slice. The simulated isotopic ratios are evaluated against Greenland ice core data. Such a dynamical downscaling of global isotope simulations for Greenland under mid-Holocene conditions is performed for the first time in the framework of this study.

## 2 Methods

**2.1 COSMO_iso**

### 2.1.1 Model Description

In this study, simulated stable water isotope concentrations of $HD^{16}O$ and $H_2^{18}O$ with isotope-enabled GCMs (section 2.1.2), are regionally downscaled with COSMO_iso (Pfahl et al., 2012), an isotope-enabled version of the numerical weather prediction model COSMO (Consortium for Small-scale Modeling; Baldauf et al., 2011) (version 4.18). For the purpose of

long-term climate simulations, isotope-routines of COSMO_iso were implemented in COSMO-CLM (Rockel et al., 2008), the climate version of COSMO. In this context, the $\delta D$ and $\delta^{18}O$ ratios in the soil water and the surface layer snow are simulated with TERRA_iso V.1 (Dütsch, 2017; Christner et al., 2018), the isotope-enabled version of the multi-layer Land Surface Model TERRA-ML (Schrodin and Heise, 2001) in COSMO. In several studies, COSMO_iso and TERRA_iso were successfully employed for the simulation of isotopic ratios in the mid-latitudes (Pfahl et al., 2012; Aemisegger et al., 2015;

Christner et al., 2018). In the present study, the model system will be applied to the Arctic region. For this, some additional modifications regarding the treatment of snow and ice had to be implemented in the model:

**Snow albedo**

The surface albedo of fresh snow is increased from 0.7 to 0.8 to improve the model agreement with measured values of short-wave reflectance and 2m temperature at stations from the Cooperative Institute for Research in Environmental Sciences at the University of Colorado Boulder (CIRES) in Central Greenland (Karremann and Schädler, 2021).

**Snow layer thickness**

In the standard configuration of COSMO, the Greenland ice sheet is treated as a constant mass of ice, which is covered by a single snow layer. But in this model structure, dynamical processes within the ice sheet (flow, basal melt) are not included. As a result, the depth of the snow layer is constantly increasing and thus also its heat capacity. To avoid this spurious model behaviour, the snow layer depth is limited to 5 cm in this study. Using this value, realistic diurnal cycles of the 2 m air temperature could be simulated.

**Marine regions with sea ice cover**

To be able to simulate reasonable fractionation processes for marine regions with sea ice cover, a snow layer is also implemented on top of the sea ice (e.g., as suggested in Bonne et al., 2019). The isotopic composition of this surface snow layer is in this case set to the isotopic composition of the most recent precipitation.

**Fractionation at snow covered surfaces**

Isotope fractionation during sublimation from a surface snow layer is poorly understood. Several different processes are suggested to be involved, which are not yet taken into account in state-of-the-art isotope enabled models (see e.g. discussion in Christner et al., 2017), such as non-fractionating layer-by-layer sublimation (e.g. Ambach et al., 1968), kinetic fractionation during sublimation into sub-saturated air, a diurnal cycle of sublimation combined with fractionating vapor deposition on the snow (e.g. Steen-Larsen et al., 2014), and fractionating melt water evaporation combined with recrystallization of residual melt water (Gurney and Lawrence, 2004). To approximate this complex interplay of different influencing factors, in this study, an equilibrium fractionation during sublimation from surface layer snow and sea ice is assumed. However, the authors are aware that this is a simplified description of isotope fractionation during sublimation.

**2.1.2 Model Simulation Setup**

The capability of COSMO_iso to realistically reproduce the fractionating processes of stable water isotopes in Greenland is evaluated. For this, the nudged simulation outputs (standard and isotopic) from an isotope-enabled atmospheric model ECHAM5-wiso (Werner et al., 2011) simulation are dynamically downscaled with COSMO_iso for the whole Arctic region. The data from the same ECHAM5-wiso simulation have been already used as boundary conditions for COSMO_iso simulations over Europe by Christner et al. (2018). The simulation outputs from ECHAM5-wiso are at a T106 horizontal

spatial resolution (1.1° x 1.1°) and on 31 atmospheric vertical levels. The dynamical fields were nudged every 6 hours towards ERA-Interim reanalysis data (Dee et al., 2011). Monthly varying sea surface temperatures and sea ice cover were prescribed as lower boundaries over sea, also based on the ERA-Interim data. The simulation period is 2008-2014. In order to guarantee at the boundaries of the regional model domain a physically consistent transition between the coarse model resolution of the GCM and the fine model resolution of the RCM, the model resolution is step-wise increased. This

procedure is called nesting. In a first nesting step, the spatial resolution of COSMO_iso is set to 0.44° x 0.44°, corresponding to 50 km x 50 km in rotated coordinates (COSMO_iso_50km). In a second nesting step, an additional COSMO_iso simulation with a spatial resolution of 0.0625° x 0.0625° (corresponding to about 7 km × 7 km) for Greenland (COSMO_iso_7km) is nested in the COSMO_iso_50km simulation. This high-resolution simulation covers the year 2011. In the COSMO_iso runs, the horizontal wind fields above the 850 hPa level are spectrally nudged (von Storch et al., 2000)

towards the reanalysis-based dynamical fields of ECHAM5-wiso. This method ensures that consistent atmospheric boundary conditions build the framework for the fractionating processes simulated in COSMO_iso. The model domains of both simulations is shown in Figure 1.

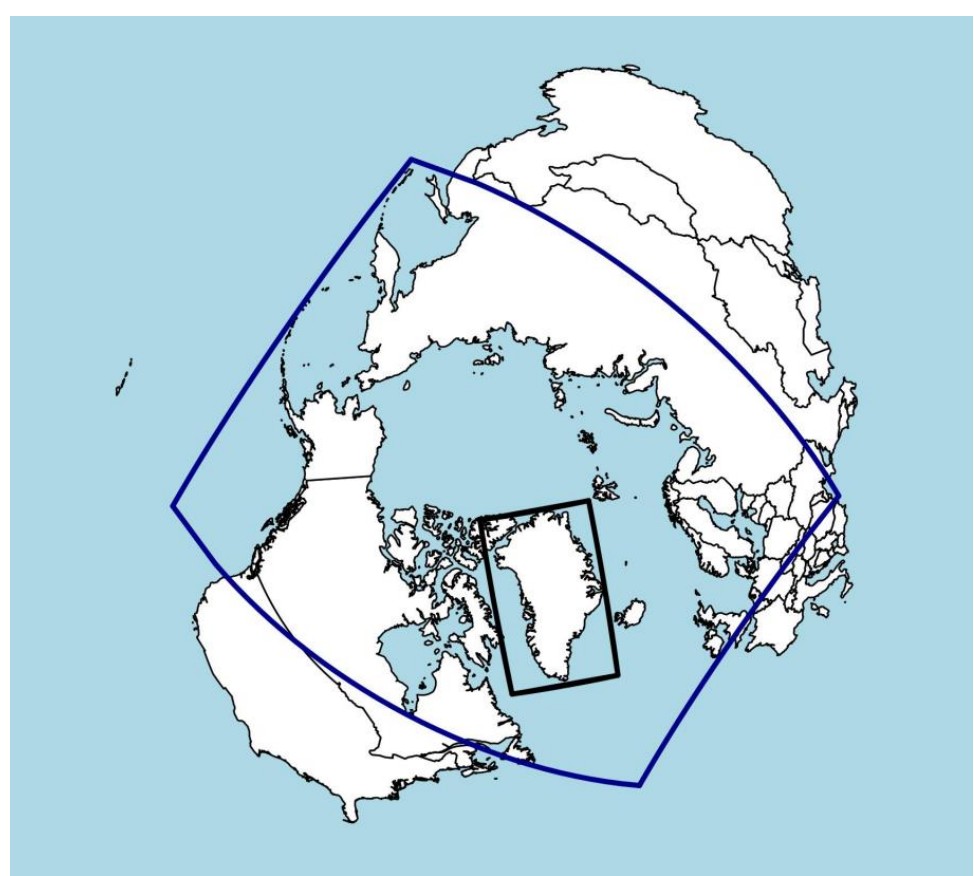

Figure 1: COSMO_iso model domain of the 50 km (blue) and the 7 km (black) simulation.

The same model chain is applied to the mid-Holocene period. Atmospheric fields have been retrieved from a mid-Holocene simulation of the fully-coupled isotope-enabled Max Planck Institute Earth System Model (MPI-ESM-wiso, Cauquoin et al., 2019), whose atmospheric component is ECHAM6-wiso. The major ECHAM6 model changes compared to ECHAM5 include an improved representation of radiative transfer in the solar part of the spectrum, an improved representation of surface albedo, a new aerosol climatology and an improved representation of the middle atmosphere (Stevens et al., 2013). The ocean component is the Max Planck Institute Ocean-Model (MPIOM, Jungclaus et al., 2013). With COSMO_iso, a representative time slice of 30 years is simulated for this climate period, only, since the regional COSMO_iso simulations are computationally very expensive. The greenhouse gas concentrations and the orbital parameters are adapted, according to the Paleoclimate Modelling Intercomparison Project 4 experiment design (PMIP4, Kageyama et al., 2018). The model domain of the COSMO_iso simulations is identical to the present-day simulations.

## 2.2 Observations

The capability of the isotope-enabled regional climate model COSMO_iso to reproduce measured isotopic ratios in Greenland is evaluated by comparing the simulation results to observational data. The simulated isotopic composition in precipitation is assessed by comparing the model results in the arctic region with observed monthly data from the Global Network of Isotopes in Precipitation (GNIP) of the International Atomic Energy Agency and the World Meteorology Organization [IAEA/WMO, 2016] over the period 2008-2014 (Table 1). Furthermore, simulated $\delta^{18}O$ ratios are compared to snow pit samples collected during the North Greenland Traverse (Fischer et al., 1998; Weißbach et al., 2016a) and top core samples from five ice core locations (Renland (Vinther et al., 2008), Neem (Masson-Delmotte et al., 2015), GISP2 (Grootes and Stuiver, 1997), Summit (Fischer, 2003), SE-Dome (Furukawa et al., 2017)). The station numbers assigned to the respective samples within this study, as well as their locations and $\delta^{18}O$ values are summarized in Table 2. Since all snow pit samples cover different time periods, the present-day $\delta^{18}O$ values (black numbers in Table 2) are calculated as an average of all available $\delta^{18}O$ values measured between 1940 and 2014. With this procedure uncertainties in snow pit samples and top ice core samples, associated with post depositional diffusion and wind erosion and the resulting constraints in analysing annual and interannual top ice core data (e.g. Johnson et al., 2000), can be neglected. However, further uncertainties in snow pit samples and ice core data remain, regarding the timescale assignment (Steig et al., 2005) and the spatial variability (Weißbach et al., 2016b).

 **Table 1:** List of GNIP stations used in this study

| No. | Station Name | Longitude | Latitude | Time period |
|-----|--------------|-----------|----------|-------------|
| 1 | Danmarkshavn | -18.66 | 76.76 | 2008-2014 |
| 2 | Ny Alesund | -11.93 | 78.91 | 2008-2014 |
| 3 | Reykjavik | -21.93 | 64.13 | 2008-2014 |
| 4 | Espoo | 24.83 | 60.18 | 2008-2010 |
| 5 | Kuopio | 27.62 | 62.89 | 2008-2010 |
| 6 | Rovaniemi | 25.75 | 66.49 | 2008-2010 |
| 7 | Snare Rapids | -116.00 | 63.52 | 2008-2010 |
| 8 | Tartu | 26.46 | 58.26 | 2013-2014 |
| 9 | Vilsandi | 21.81 | 58.38 | 2013-2014 |

**Table 2:** Description of the snow pit and ice core samples used in this study. The present-day $\delta^{18}O$ values are calculated as an average of all available $\delta^{18}O$ values measured in snow pit samples between 1940-2014. The mid-Holocene $\delta^{18}O$ values are calculated as an average of the measured $\delta^{18}O$ values in ice cores over the period 5.5 ka – 6.5 ka. Black numbers indicate present-day $\delta^{18}O$ values and the respective standard deviations, blue numbers in parentheses indicate mid-Holocene values and the corresponding standard deviations.

| No. | Name | Sample | Longitude | Latitude | δ18O | σ | Reference |
|-----|------|--------|-----------|----------|------|---|-----------|
| 1 | Renland | top core | -26,73 | 71,27 | -27.38 (-26.44) | 0.42 (0.31) | Vinther et al., 2008 |
| 2 | NEEM | top core | -51,06 | 77,45 | -33.24 | 1.15 | Masson-Delmotte et al., 2015 |
| 3 | GISP2 | top core | -38,48 | 72,58 | -34.95 (-34.83) | 0.69 (0.33) | Grootes & Stuiver, 1997 |
| 4 | Summit | top core | -37,64 | 73,03 | -36,46 | 1,11 | Fischer, 2003 |
| 5 | B27_B28 | snow pit | -46,48 | 76,65 | -34,05 | 1,52 | Weißbach et al., 2016 |
| 6 | NGT03C93 | snow pit | -37,62 | 73,94 | -37,02 | 1,06 | Weißbach et al., 2016 |
| 7 | NGT06C93 | snow pit | -37,62 | 75,25 | -36,89 | 1,38 | Weißbach et al., 2017 |
| 8 | NGT14C93 | snow pit | -36,4 | 76,61 | -36,18 | 1,64 | Weißbach et al., 2017 |
| 9 | NGT23C94 | snow pit | -36,5 | 78,83 | -35,18 | 1,78 | Weißbach et al., 2018 |
| 10 | NGT27C94 | snow pit | -41,13 | 80 | -34,01 | 1,55 | Weißbach et al., 2018 |
| 11 | NGT30C94 | snow pit | -45,91 | 79,34 | -34,19 | 1,69 | Weißbach et al., 2019 |
| 12 | NGT33C94 | snow pit | -44 | 78 | -36,13 | 1,62 | Weißbach et al., 2019 |
| 13 | NGT37C95 | snow pit | -49,21 | 77,25 | -33,81 | 1,38 | Weißbach et al., 2020 |
| 14 | NGT39C95 | snow pit | -46,48 | 76,65 | -34,95 | 1,4 | Weißbach et al., 2020 |
| 15 | NGT42C95 | snow pit | -43,49 | 76 | -35,53 | 1,24 | Weißbach et al., 2021 |
| 16 | NGT45C95 | snow pit | -42 | 75 | -35,33 | 1,32 | Weißbach et al., 2021 |
| 17 | GRIP | top core | -37,64 | 72,58 | -34,73 | 0,32 | Vinther et al., 2006 |
| 18 | NGRIP | top core | -42,32 | 75,1 | -34,69 | 0,36 | Vinther et al., 2006 |

Since both snow pit samples and top core samples from ice cores represent an integrated signal of the isotopic composition in precipitation, we compute modeled annual mean $\delta^{18}O$ values in precipitation and compared the multi-year 2008-2014 model mean to the observed values. For the calculation of the yearly modelled mean values, the modelled $\delta^{18}O$ in precipitation is weighted with accumulation rate, i.e. months with high precipitation amounts get a higher weight compared to months with small precipitation amounts.

Ice core samples are also used to evaluate the simulated isotopic ratios for the mid-Holocene. Beside the already mentioned Renland and GISP2 samples, two more ice core samples, namely GRIP and NGRIP (Vinther et al., 2006), are used for the model evaluation. The mid-Holocene $\delta^{18}O$ values (blue numbers in Table 2) are calculated as an average of the measured $\delta^{18}O$ values in ice cores over the period 5.5 ka – 6.5 ka.

## 3 Results

### 3.1 Present-Day

#### 3.1.1 Standard climatological parameter

In a first step, the general capability of the COSMO model to reproduce observed standard climatological parameter in present-day simulations for Greenland is assessed. For this purpose, the results of an ERA-Interim reanalysis (Dee et al., 2011) driven simulation with the standard COSMO model (without isotope application) are compared with observations, collected by the Danish Meteorological Institute (DMI). Evaluated are the yearly mean 2 m temperatures (Figure 2a) and the yearly mean precipitation sums (Figure 2b) over the period 1995 – 2015. Both simulated 2 m temperatures as well as simulated precipitation amounts are in good agreement with the DMI observations. Especially the simulated 2 m temperatures coincide well with the observed values. For the precipitation sums, the spread of simulated and observed values is higher than for the 2 m temperatures, a feature generally occurring in weather and climate simulations. Thus, the COSMO model is generally able to simulate reasonable climate conditions in Greenland and can therefore be used for isotope applications in this region. A detailed analysis of the standard COSMO model performance in the Arctic region is presented in Karremann and Schädler, (2021).

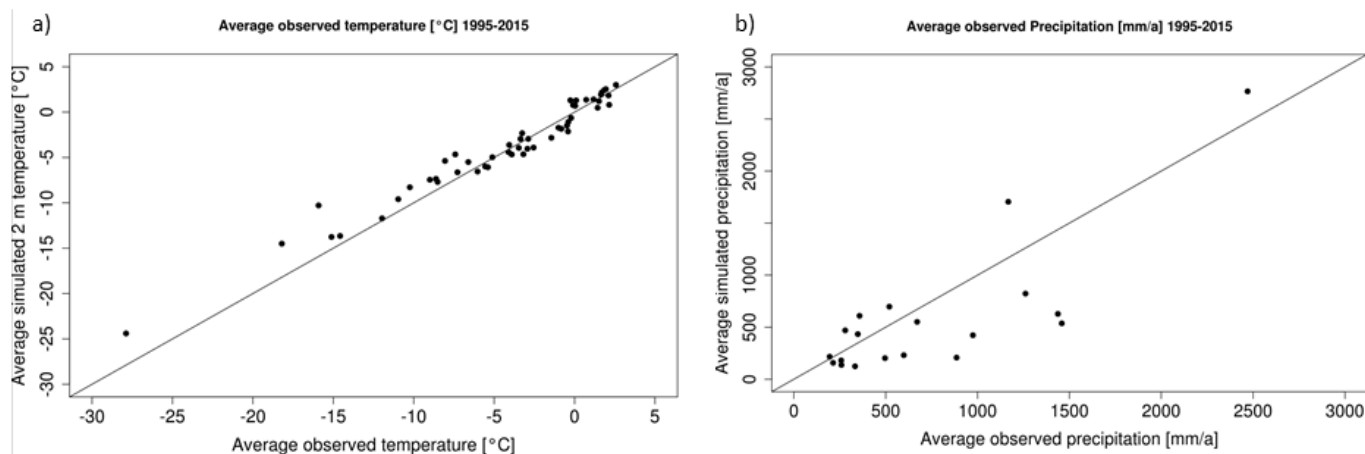

Figure 2: Simulated yearly mean (a) 2 m temperatures and (b) precipitation sums of a standard COSMO simulation, driven with ERA-Interim, for Greenland over the period 1995-2015 compared to DMI observations. The black solid line is the 1:1 line in (a) and (b).

### 3.1.2 Comparison of simulated $\delta^{18}$O data to station data

Figure 3 shows the yearly mean $\delta^{18}$O values for the period 2008-2014 for Greenland, simulated with COSMO_iso_50km (a) and ECHAM5-wiso (b). Additionally, the locations and the observed $\delta^{18}$O values of the 19 snow pit samples, used to assess the models' capability to reproduce observed $\delta^{18}$O ratios in Greenland, are illustrated. In general, COSMO_iso in a 50 km x 50 km spatial resolution is able to reproduce the observed isotopic ratios at the snow pit samples and improves the simulation results of ECHAM5-wiso. In both simulations, the $\delta^{18}$O ratios are high near the coastline and low in Central Greenland. But

in COSMO_iso_50km, the $\delta^{18}$O ratios decline more rapidly from the coastline to the inland plateau than in ECHAM5-wiso. The spatial $\delta^{18}$O differences are consequently more pronounced and the general overestimation of $\delta^{18}$O ratios, which occurs in ECHAM5-wiso, is reduced in COSMO_iso_50km. As a consequence, the regional simulation reaches a better agreement with the observations. In COSMO_iso_50km the root mean squared error (RMSE) is reduced by 0.98 ‰ over all snow pit samples (the RMSE of ECHAM5-wiso is 2.42 ‰ and the RMSE of COSMO_iso_50km is 1.44 ‰). This RMSE reduction is

significant at the 95 % level, assessed by performing a t-test. Especially for the snow pit samples for which ECHAM5-wiso exhibits strong deviations from the observed $\delta^{18}$O values (1,3,4,6,7,8,9,10,16,17,19; see Table 2), a regional downscaling with COSMO_iso_50km reduces the bias considerably (Figure 4). For these stations, the RMSE of ECHAM5-wiso of 3.09 ‰ is reduced by 1.65 ‰ to 1.44 ‰. But for snow pit samples at which ECHAM5-wiso has already a high agreement with the observations (2,5,11,13,14), COSMO_iso_50km increases the RMSE from 0.34 ‰ to 1.51 ‰.

Figure 5 shows that these annual biases of the COSMO_iso_50km simulation are not caused by systematic seasonal biases, as for example reported by Sjolte et al., (2011) for RCM simulations in Greenland. Shown are the simulated monthly $\delta^{18}$O values with COSMO_iso_50km compared to observed monthly $\delta^{18}$O values in precipitation for the period 2008-2014, collected at arctic stations of the GNIP dataset (Table 1). In general, the modelled $\delta^{18}$O values in precipitation are in good

agreement with the monthly GNIP data. But in the COSMO_iso_50km simulation no systematic over- or underestimation of observed isotope ratios is simulated with the RCM. This is true for each season. Neither in winter (low $\delta^{18}$O values), nor in summer (high $\delta^{18}$O values), systematic deviations to the observations are simulated. Thus, the seasonal variability in the COSMO_iso_50km results has no systematic impact on the yearly mean $\delta^{18}$O values and is therefore not the reason for systematic differences between the coarse model results and observations.

As visible in Figure 3, these systematic differences are rather caused by a southward shift of the area of low yearly mean $\delta^{18}$O values in central Northern Greenland in COSMO_iso_50km relative to ECHAM5-wiso. As a result, the simulated $\delta^{18}$O values in central Northern Greenland in COSMO_iso_50km are higher than in ECHAM5-wiso. Since there, ECHAM5-wiso has already a high agreement with the observed $\delta^{18}$O values, a model bias is introduced in COSMO_iso_50km, causing the deviations relative to the observations in Northern Greenland. But in all, COSMO_iso_50km yields an overall improvement in simulating the yearly mean $\delta^{18}$O values compared to ECHAM5-wiso.

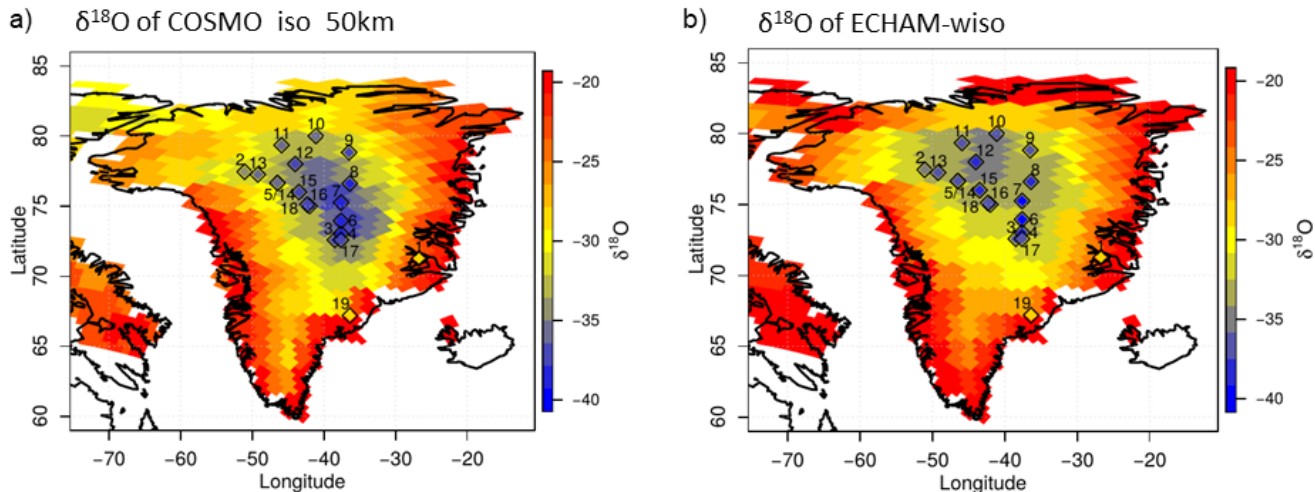

**Figure 3:** Yearly mean $\delta^{18}$O values of COSMO_iso_50km (a) and ECHAM5-wiso (b, interpolated to the COSMO_iso_50km grid) for the period 2008 - 2014 and the corresponding observations for the 19 snow pit samples (Table 2).

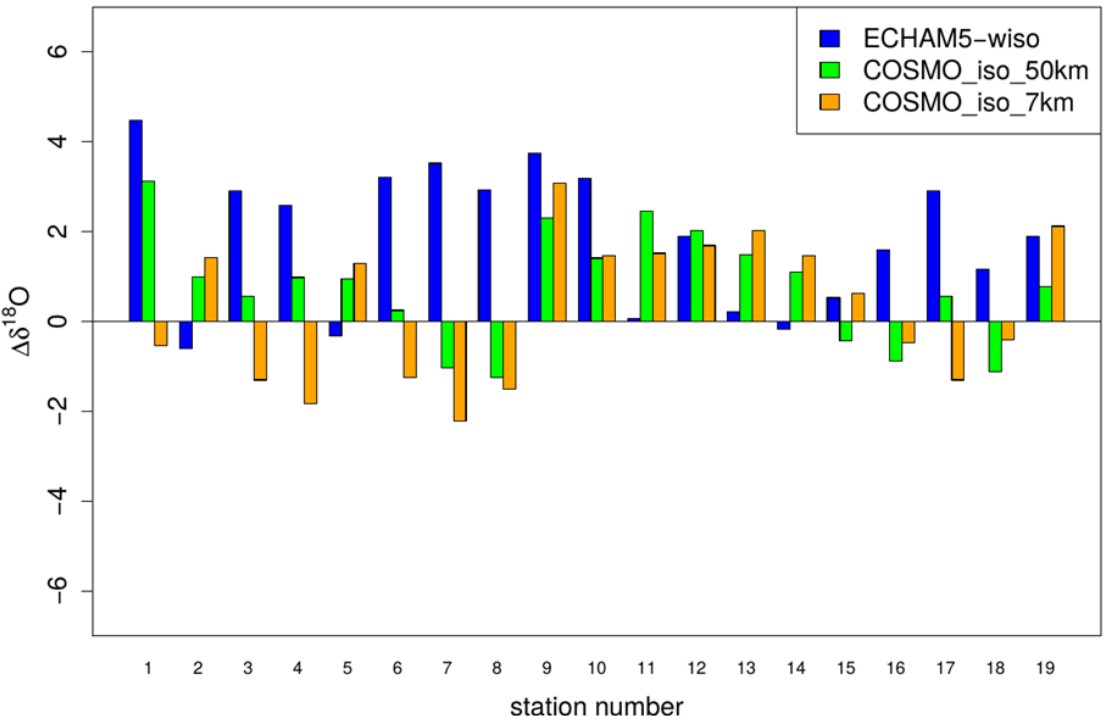

**Figure 4:** Differences (Δ) between simulated and observed δ¹⁸O values (model minus observation) for the model results of ECHAM5-wiso, COSMO_iso_50km, and COSMO_iso_7km, and snow pit samples / top core samples from ice cores from Greenland (simulation: 2011, observation: mean present values). Numbers refer to the different snow pit locations shown in Figure 3.

A further downscaling with COSMO_iso to a spatial resolution of 7 km x 7 km does not improve the simulation results further. The RMSE reduction in comparison to ECHAM5-wiso is 0.83 ‰ (compared to 0.98 ‰ for COSMO_iso_50km). The only exception constitutes the snow pit sample from Renland (1). Here, a considerable model bias in ECHAM5-wiso and COSMO_iso_50km is strongly reduced in COSMO_iso_7km. The coastal area of Renland is characterized by complex terrain and constitutes a special case for isotope-enabled modeling in Greenland. The snow pit sample is located in a transition zone from the homogeneous inland glaciation to the rugged coastline, where the glaciers calve into the sea. Thus, within short distances large differences in altitude and land surface characteristics occur in this region. The isotopic ratios in the snow pit sample are therefore strongly affected by these heterogeneous local conditions, which are insufficiently represented in the coarse model resolution of ECHAM5-wiso. By increasing the spatial resolution with regional climate modelling, also the representation of the associated small-scale processes is improved. This leads generally to an improved agreement of the simulation results with observations, as seen for the COSMO_iso_7km run for Renland (Figure 4).

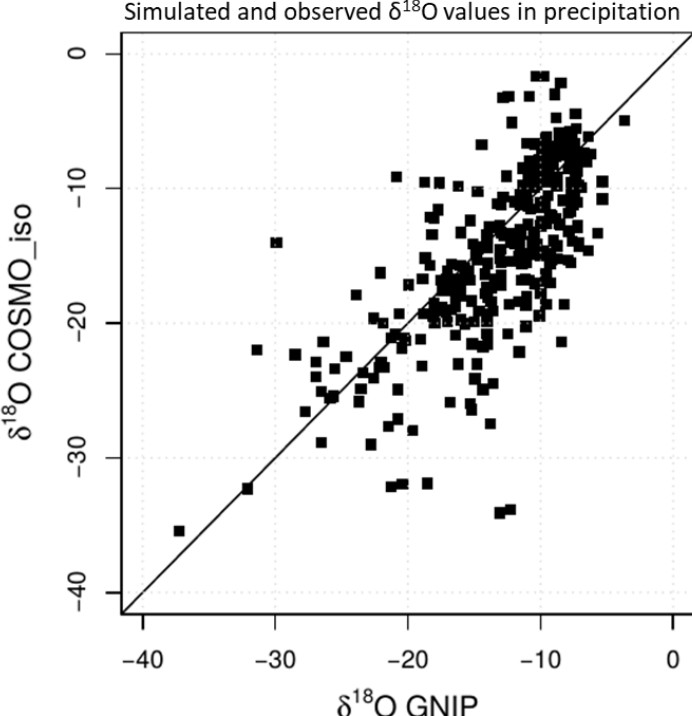

**Figure 5:** Monthly δ¹⁸O simulated with COSMO_iso_50km for the period 2008 - 2014 and the corresponding observations for 9 GNIP stations (Table 1).

However, an increase in spatial resolution is also associated with an increased heterogeneity of the surface characteristics and the related small-scale processes, especially in complex terrains. This is because the GCM grid boxes are further divided in smaller RCM grid boxes and consequently higher as well as lower values (for e.g. altitude) are now included in the respective GCM grid boxes. As a consequence, an additional spatial variability is introduced in the RCM simulations in comparison to the GCM results. Due to uncertainties accompanied by model simulations, this can potentially increase the

RCM bias with respect to in situ point measurements, which may actually be closer to the spatially averaged values simulated by the coarse GCM model. This effect can be observed for the SE-Dome ice core (19) in southeastern Greenland. Comparable to the Renland ice core, SE-Dome is located near the coastline. But in contrast to the Renland ice core, an increase in the spatial resolution to 7 km does not further improve the RCM results for SE-Dome. On the contrary, the δ¹⁸O bias is even higher than in the ECHAM5-wiso simulation.

Despite the lack of improvement in the point to grid-cell comparison, higher resolved RCM simulations allow the subgrid-scale variability of δ¹⁸O within GCM grid boxes to be simulated and compared to observed δ¹⁸O values. In this way, the inherent uncertainty of in situ measurements, associated with a local micrometeorological variability, can be considered. Thus, in the following sections, snow pit samples are no longer solely compared to the model grid boxes covering the

samples location. Instead, it is investigated whether the $\delta^{18}O$ range of all adjacent RCM grid boxes to a snow pit location is
consistent with the observed $\delta^{18}O$ value of the same site. For this, all RCM grid boxes located within the corresponding GCM grid box are included in the comparison with the observations.

### 3.1.3 $\delta^{18}O$ variability

The spatial isotopic ratio variability of the COSMO_iso_50km grid boxes surrounding the 19 snow pit samples is shown as a
Box-Whiskers-plot in Figure 6a. The spatial isotopic ratio variability of the COSMO_iso_7km is shown in Figure 6b. In this spatial isotopic ratio variability, the $\delta^{18}O$ values of all COSMO_iso (50 km) and COSMO_iso (7 km) grid boxes within the ECHAM5_wiso grid box closest to the snow pit sample, are included. For 14 of the 19 snow pit samples (1,2,3,4,5,6,7,8,14,15,16,17,18,19) the observed $\delta^{18}O$ values are within the range of the spatial COSMO_iso_50km grid box variability. But for 5 of the 19 snow pit samples (9-13) the spatial isotope range of the COSMO_iso_50km simulation does
not fit with the observations. Since these stations are all located in the north of Greenland (Figure 3), this is likely partially associated with the southward shift of the area of low yearly mean $\delta^{18}O$ values in central Northern Greenland in COSMO_iso_50km in comparison to ECHAM5-wiso, as already described in section 3.1.2.

A downscaling to 7 km does slightly increase the spread of the COSMO_iso results. But still, the observed $\delta^{18}O$ values from 5 of 19 snow pit samples fall outside the range of the modelled COSMO_iso_7km grid box variability (Figure 6b). Thus, a
further downscaling to a spatial resolution of 7 km does not substantially change the simulated isotopic ratio spread within an ECHAM5-wiso grid box. In accordance with the missing benefits of the COSMO_iso_7km simulation and its increased computing time costs, only a COSMO_iso_50km simulation is performed for the mid-Holocene (section 3.2).

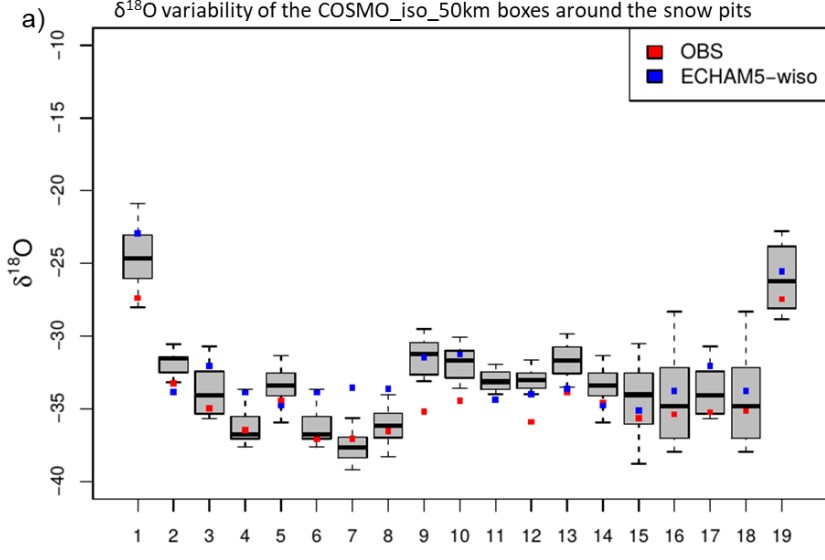

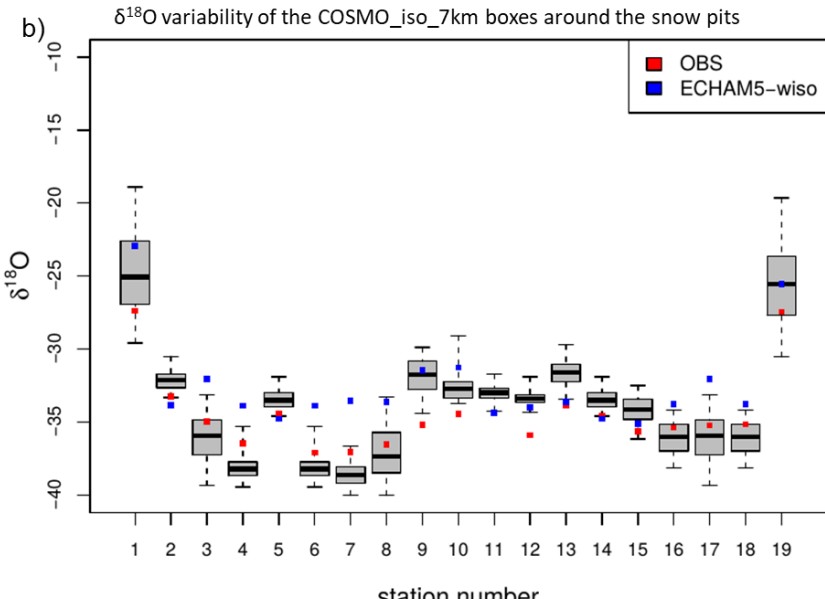

**Figure 6:** Present-day isotopic ratio variability of the COSMO_iso grid boxes, surrounding the 16 snow pit samples for the (a) 50 km and (b) 7 km simulation. The black bar in the Box-Whiskers-plot represents the median of the isotope ratio distribution. The box comprises the upper and lower quartile, the whiskers the whole distribution. The MPI-ESM-wiso results are shown by the blue dots and the observed $\delta^{18}O$ values are shown by the red dots.

The high spatial $\delta^{18}O$ variability in COSMO_iso simulations is also reflected in the spatial $\delta^{18}O$-temperature slope of the COSMO_iso_50km run (Figure 7c), a measure that is frequently used to analyze how strongly isotope ratios and surface temperatures are interrelated. The spatial isotope-temperature slope constitutes a linear fit between the simulated $\delta^{18}O$ ratios

and the surface temperatures at all COSMO_iso_50km grid boxes within the respective ECHAM5-wiso grid box. The spatial

$\delta^{18}$O-temperature slope is high in Central Greenland (1.5 - 2.0) and moderate at the coastline (~ 1.0). In order to better understand these spatial $\delta^{18}$O-temperature patterns, both quantities affecting the spatial $\delta^{18}$O-temperature slope, i.e. the spatial $\delta^{18}$O variability and the spatial temperature variability, are explicitly analyzed in Figure 7a and Figure 7b. These spatial variabilities are calculated as the standard deviation of all COSMO_iso_50km grid boxes within the respective ECHAM5-wiso grid boxes.

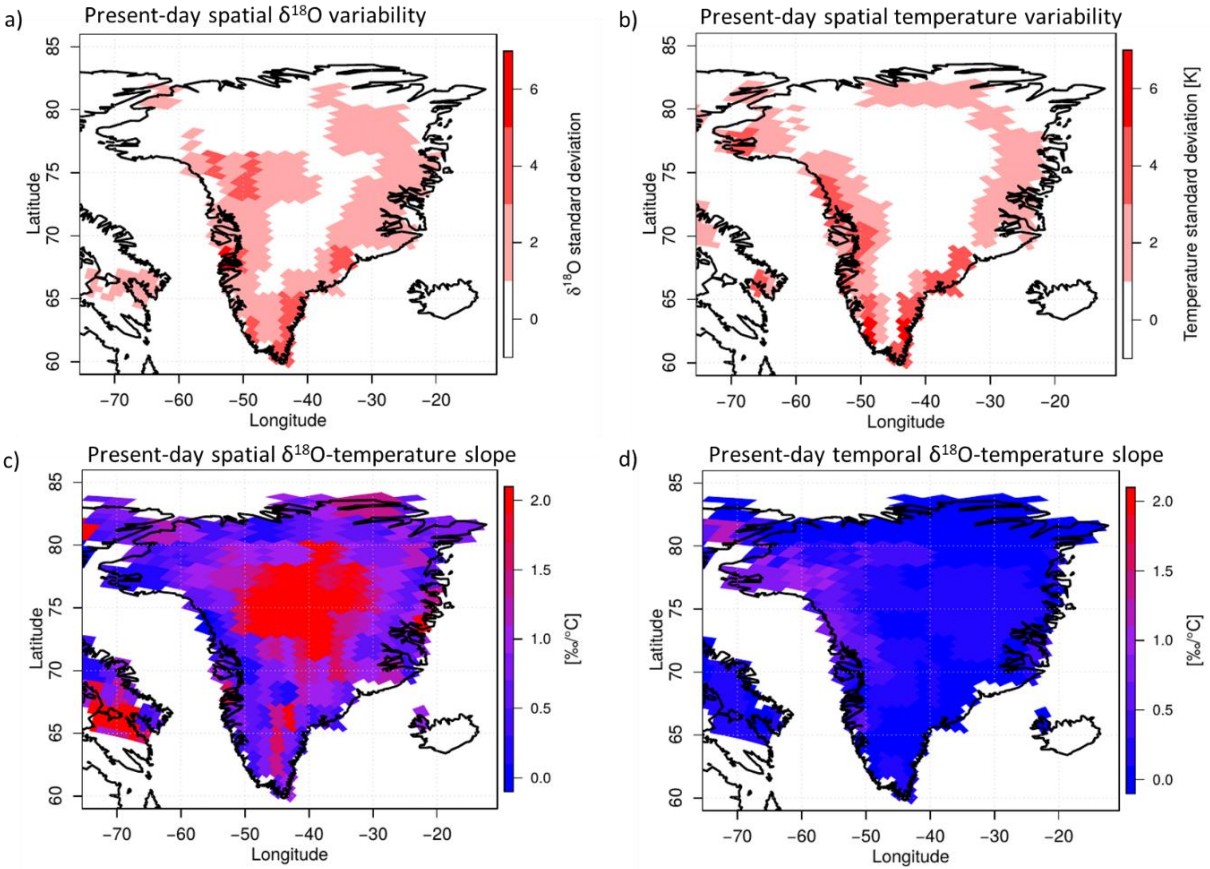

**Figure 7:** Present-day spatial subgrid-scale variability (calculated as standard deviation) of (a) $\delta^{18}$O and (b) surface temperature, derived from the COSMO_iso_50km grid boxes within the respective ECHAM5-wiso grid boxes for whole Greenland. Present-day (c) spatial and (d) interannual temporal $\delta^{18}$O-temperature slope for Greenland, based on yearly mean values.

Figure 7a shows that at the coastline, the spatial $\delta^{18}$O variability of COSMO_iso_50km is considerably increased within the ECHAM5-wiso grid boxes. In Central Greenland the spatial isotopic ratio variability is lower. Thus, the simulated spatial $\delta^{18}$O variability is high in regions where large orographic differences occur within short distances, like the coastal areas of Greenland, and lower for homogeneous terrain like the inland plateau. Nevertheless, widespread areas with higher spatial isotopic variability occur also in the inland plateau of Greenland. This is not the case for the spatial surface temperature

variability. In Central Greenland, surface temperature variability is very low (Figure 7b). But near the coastline the spatial surface temperature variability is also high, highlighting how important the land surface characteristics are for the regional temperature variability. The moderate spatial $\delta^{18}$O-temperature slope at the coastline (Figure 7c) is therefore a result of a high surface temperature variability in this region counteracting the high $\delta^{18}$O variability in the slope calculation. There, the correlation between both quantities is consequently high (Figure S1a in Supplementary Material, mainly between 0.7 and

0.99). In Central Greenland, the spatial $\delta^{18}$O-temperature slope is further increased due to the relatively high $\delta^{18}$O variability compared with the surface temperature variability. Therefore, the spatial distribution of $\delta^{18}$O cannot be solely explained by land surface processes and the associated spatial temperature variability. The additional spatial $\delta^{18}$O variability consequently must be caused by dynamic atmospheric processes. In this way, isotopic ratios based on atmospheric fractionation processes along the trajectory of an air mass are transported to Central Greenland and increase the isotopic variability there.

In order to investigate the temporal interrelations between the isotope ratios and the surface temperature, the interannual temporal $\delta^{18}$O-temperature slope is calculated for the COSMO_iso_50km simulation, based on the yearly mean $\delta^{18}$O and surface temperature values (Figure 7d). The interannual temporal $\delta^{18}$O-temperature slope is, in contrast to the spatial $\delta^{18}$O-temperature slope, small all over Greenland, which is in agreement with the results of Sjolte et al., (2011). That means that the interannual $\delta^{18}$O variability is less pronounced than the interannual surface temperature variability and thus, also the

correlation between both quantities is not as strong as for the spatial $\delta^{18}$O variability and the spatial temperature variability (Figure S1b in Supplementary Material). The mean spatial correlation over Greenland is 0.61 compared to 0.25 for the mean temporal correlation. The impact of interannual surface temperature variations on the temporal $\delta^{18}$O variability in Greenland is therefore not as dominant as for the spatial $\delta^{18}$O variability.

## 3.2 Mid-Holocene

### 3.2.1 Comparison of simulated $\delta^{18}$O data to ice core data

In contrast to the present-day simulations, for the mid-Holocene, COSMO_iso_50km is driven by MPI-ESM-wiso rather than COSMO_iso_50km. While in ECHAM5-wiso oceanic boundary conditions are prescribed by monthly varying sea surface temperatures and sea ice cover, ocean states are calculated internally in the fully-coupled atmosphere-ocean Earth-

365 System-Model MPI-ESM-wiso. Systematic deviations between the COSMO_iso_50km simulations for mid-Holocene and present-day, caused by these different forcing approaches, therefore, cannot be excluded. For this reason, a comparison of the mid-Holocene $\delta^{18}$O anomalies to the present-day conditions is omitted and an analysis is performed for simulated Mid-Holocene $\delta^{18}$O ratios with comparison to observed Mid-Holocene $\delta^{18}$O values.

In Figure 8a the absolute differences between the simulated MPI-ESM-wiso (blue) and COSMO_iso_50km (green) grid box

results and the observed $\delta^{18}$O ratios at the corresponding ice cores are presented for the mid-Holocene. As in Figure 6, the spatial isotopic ratio variability of the COSMO_iso_50km grid boxes surrounding four Greenland ice core samples is shown as a Box-Whiskers-plot. MPI-ESM-wiso properly reflects the isotopic ratios of the mid-Holocene from ice core data. For the

inland ice cores (GRIP, GISP2, NGRIP), the simulated $\delta^{18}O$ deviates only by about 1 ‰ relative to the observation, at Renland the deviation is about 3 ‰. For GRIP and GISP2 the MPI-ESM-wiso simulations slightly underestimate the $\delta^{18}O$ ratios, for NGRIP and Renland, the $\delta^{18}O$ values are slightly overestimated.

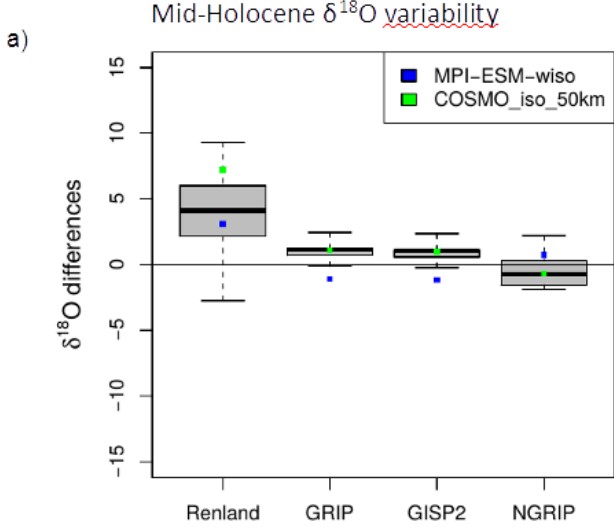

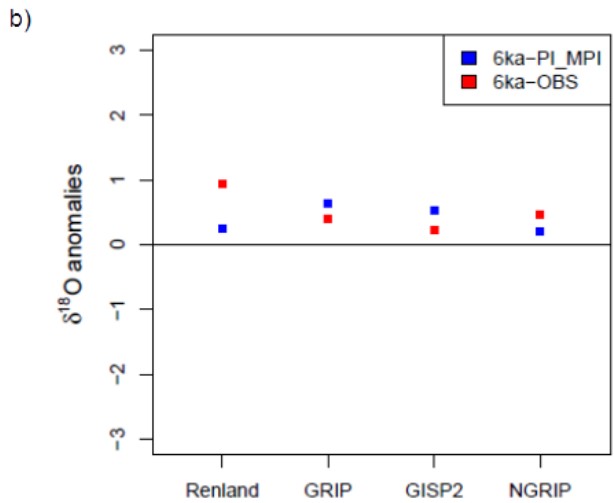

**Figure 8:** (a) mid-Holocene isotopic ratio variability of the COSMO_iso_50km grid boxes surrounding four Greenland ice core samples. In each grid box, the observed ratios derived from the ice cores are subtracted from the simulated $\delta^{18}O$ ratios. The black bar in the Box-Whiskers-plot represents the median of the isotope ratio distribution. The box comprises the upper and lower quartile, the whiskers the whole distribution. The MPI-ESM-wiso (blue dots) and COSMO_iso_50km (green dots) results for the grid points closest to the ice cores are also shown as differences with respect to the observed $\delta^{18}O$ ratios. (b) anomalies of the MPI-ESM-wiso simulation relative to pre-

industrial (PI) conditions, based on an MPI-ESM-wiso PI-reference simulation (Cauquoin et al., 2019) are shown as blue points, the observed anomalies for the mid-Holocene relative to present-day are shown as red points.

For COSMO_iso_50km, the deviation of $\delta^{18}O$ values relative to the observations are opposite in sign compared with MPI-ESM-wiso at all locations except Renland. That means that in GRIP and GISP2, the underestimated $\delta^{18}O$ values in MPI-ESM-wiso are turned into overestimated $\delta^{18}O$ values in COSMO_iso_50km, at NGRIP the overestimation is turned into an underestimation, but the net bias is not reduced. At Renland, the bias is even increased. Thus, by just looking at the absolute

biases, the downscaling does not seem to bring an added value to the MPI-ESM-wiso results for mid-Holocene conditions. However, when the spatial isotopic ratio variability within MPI-ESM-wiso grid cells simulated by COSMO_iso_50km isotopic ratios is taken into account, the model results are in agreement with the isotopic ratios of the ice core samples.

In Figure 8b, the MPI-ESM-wiso model anomalies with reference to the pre-industrial period (PI) conditions, which are based on an MPI-ESM-wiso PI-reference simulation performed by Cauquoin et al. (2019), are compared to the measured

mid-Holocene-PI $\delta^{18}O$ anomalies of the ice cores. The positive $\delta^{18}O$ anomalies between mid-Holocene and PI for both ice core data and MPI-ESM-wiso model results are associated with higher temperatures, especially during the summer, and a reduction in Arctic sea-ice during mid-Holocene (Cauquoin et al., 2019). In Renland and NGRIP simulated anomalies are slightly underestimated, and in GRIP and GISP2 anomalies are slightly overestimated. But overall, the biases of the MPI-ESM-wiso mid-Holocene-PI model anomalies to the observed mid-Holocene-PI anomalies are for all ice cores small and

statistically not significant at the 95 % level (assessed by performing a t-test).

### 3.2.2 $\delta^{18}O$ variability

The fact that, in contrast to the present-day simulations, only four observational data sets are available for the mid-Holocene, makes the assessment of the simulation results difficult. Moreover, the GRIP and GISP2 ice cores being located very close to

each other (Figure 9), only three local isotope distributions clearly different from each other are available. Therefore, in Figure 9a, the spatial $\delta^{18}O$ variability of the COSMO_iso_50km simulation for the Mid-Holocenen is illustrated for all of Greenland, which is, in accordance to the analysis of the present-day simulation, again calculated as the standard deviation of all COSMO_iso_50km grid boxes within the respective GCM grid boxes. In general, the $\delta^{18}O$ variability of COSMO_iso_50km in the mid-Holocene is high at the coastline, while it is lower in Central Greenland. The Renland ice

core is consequently located in an area of a high isotopic variability, and the GRIP and GISP2 ice cores in an area of low isotopic variability. But regions with increased isotopic variability occur also in the inland plateau of Greenland. The NGRIP ice core, for instance, is located in such an area of a moderate isotopic ratio variability. The four ice core drill sites are therefore located in three regions of Greenland with substantially different sub-grid isotopic ratio variabilities.

The spatial surface temperature variability in the COSMO_iso_50km mid-Holocene simulation is shown in Figure 9b. The

mid-Holocene simulation shows a high spatial surface temperature variability near the coastline and almost no variability in Central Greenland. As a consequence, the spatial $\delta^{18}O$-temperature slope is moderate at the coastline (~ 1.0) and high in

Central Greenland ((1.5 – 2.0; Figure 9c). Moreover, the interannual $\delta^{18}$O-temperature slope is very small over Greenland in the mid-Holocene (the mean interannual slope over Greenland is 0.47), although in some regions high temporal slopes are simulated (Figure 9d). But in principle, the influence of interannual surface temperature variations on the temporal $\delta^{18}$O

variability in the mid-Holocene is not as dominant as for the spatial variability.

In general, the results of the COSMO_iso_50km mid-Holocene simulation exhibit the same spatial characteristics as for the present-day simulation (Figure 7 and Figure 9). Comparable spatial patterns are simulated for the surface temperature variability (Figure 7b and Figure 9b) as well as the $\delta^{18}$O variability (Figure 7a and Figure 9a) within a GCM grid box, although regions of increased $\delta^{18}$O variability in Central Greenland are more widely present in the mid-Holocene run than in

the present-day one. In the inland plateau, regions of a high spatial $\delta^{18}$O-temperature slope are therefore more extensive in the mid-Holocene than under present-day conditions (Figure 7c and Figure 9c). Nevertheless, the spatial $\delta^{18}$O-temperature interrelations are in both periods comparable. This is also the case for the temporal variabilities of $\delta^{18}$O and the surface temperature in the COSMO_iso_50km simulation results for the mid-Holocene and the present-day (Figure 7d and Figure 9d), although both simulations are driven by two different forcing approaches (ECHAM5-wiso nudged to ERA-Interim

reanalysis with prescribed monthly varying oceanic boundary conditions vs. the fully-coupled atmosphere-ocean Earth-System-Model MPI-ESM-wiso). This finding indicates that the spatial and interannual $\delta^{18}$O variability of COSMO_iso_50km within a GCM grid box over Greenland is not strongly dependent of the oceanic boundary conditions.

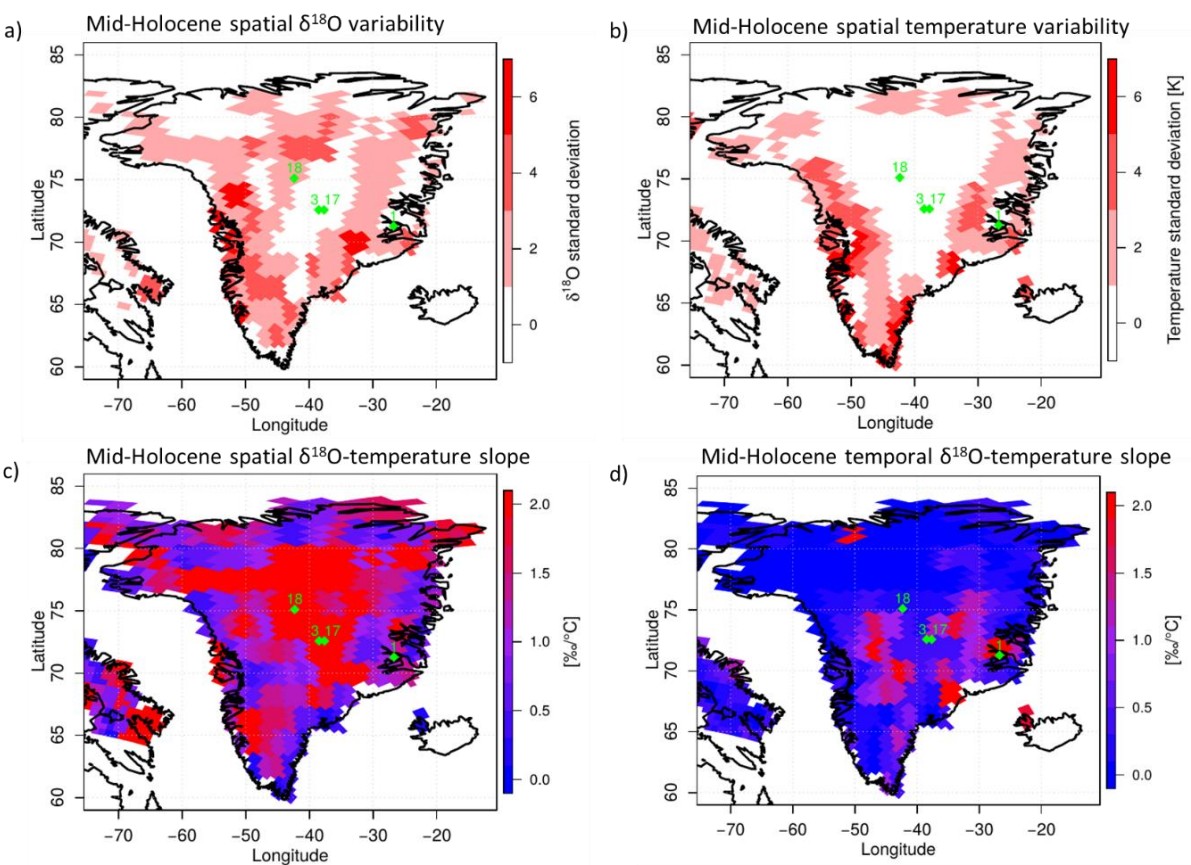

**Figure 9:** As Figure 7, but for the Mid-Holocene. The locations of the ice core samples are shown in green.

## 4 Discussion and Conclusions

The results of several global paleo-climate simulations exhibit considerable deviations from the observed regional climate patterns during the Holocene (Braconnot et al., 2012). In the presented study, for the first time, regional climate simulations with an isotope-enabled RCM are performed for Greenland to potentially improve the agreement with climate observations in this region for the mid-Holocene. In a first step, the capability of the isotope-enabled RCM COSMO_iso to reproduce observed isotopic ratios for Greenland is demonstrated.

The COSMO_iso simulation results show that a spatial resolution of 50 km produces reasonable $\delta^{18}$O values. Especially in regions where the global ECHAM5-wiso model, which has been used to derive necessary forcing fields for the COSMO_iso simulations, deviates strongly from the observed $\delta^{18}$O values, the RMSE is significantly reduced by 1.65 ‰ for regional climate simulation with COSMO_iso. In complex terrain like the coastal areas of Greenland, the results can be further improved with an additional downscaling to a spatial resolution of 7 km. In such simulations with high spatial resolution,

small-scale processes are described in more detail (e.g. Torma et al., 2015; Coppola et al., 2018) and thus the local characteristics at ice core sites are better taken into account (Sturm et al., 2005; Werner et al., 2011). But for the rest of Greenland, highly resolved regional climate simulations do not yield further improvements. For northern Greenland, regional climate simulations with COSMO_iso increase the bias with respect to observations. A comparison of simulated isotope ratios in precipitation with measured values at GNIP stations shows that such deviations between model results and observations are not caused by systematic seasonal biases in the RCM, as it was the case in a similar study by Sjolte et al., (2011) for Greenland. In central northern Greenland, rather, a model bias is introduced, due to a southward shift of the area of low yearly mean $\delta^{18}O$ values. All in all, the results of this study show that COSMO_iso is generally able to provide reasonable isotopic ratios for Greenland and the model can be applied for paleo-climate simulations.

For the mid-Holocene, MPI-ESM-wiso is in good agreement with observed ice core data in Greenland, as already described by Cauquoin et al. (2019). The model bias is, in this context, not further reduced by a downscaling with COSMO_iso. But an increase in the spatial model resolution leads also to an increase in the models' degrees of freedom. This in turn can lead to additional noise and thus, a deviating RCM behaviour and even an increase in the absolute model bias, as seen for the Renland station.

Another consequence of these increased degrees of freedom in the COSMO_iso simulation is that the spatial variability of the simulated $\delta^{18}O$ ratios is enhanced. This enhanced spatial variability represents the subgrid-scale uncertainty of the driving GCM, which can be derived in a physically consistent way by a regional downscaling. By analysing this subgrid-scale variability, the spatial uncertainties in the comparison between GCM data and point measurements can be considered. In this way, it can be demonstrated that most of the observed $\delta^{18}O$ values lie within the local $\delta^{18}O$ uncertainties of the coarse GCM results. This applies for both the present-day runs and the regional paleo-climate simulations for the mid-Holocene in Greenland. The deviation between the coarser resolved GCM results and the finer resolved observations is therefore potentially caused by the missing representation of important small-scale processes, which are induced by e.g. the surface conditions or orographic effects over Greenland. Shi et al., (2020), for instance, were able to demonstrate that GCM deficiencies in reproducing the observed water isotope variability in the southeastern Tibetan Plateau are associated with the missing representation of such small-scale processes in coarse GCM simulations.

As $\delta^{18}O$ ratios are used as an indicator for temperatures in past climates (Dansgaard et al., 1969; Masson-Delmotte et al., 2005; Jouzel, 2013), it is important to understand how the presented COSMO_iso simulations might be able to improve these isotope-based temperatures reconstructions. In general, the regional surface temperature variability and the regional $\delta^{18}O$ variability show similar patterns for Greenland. In both cases the variability is high at the coast and low on the inland plateau. Similar patterns as in the mid-Holocene can also be seen for the present-day simulations. These pattern of spatial variability in $\delta^{18}O$ and the surface temperature are in line with the results of Sjolte et al. (2011) for RCM simulations under present-day conditions for Greenland. Based on these patterns of variability, it can be derived that the regional surface temperature variability highly depends on the surface characteristics in Greenland. However, for the regional isotopic ratio variability, this dependence appears to be less pronounced. At the coastline, a clear relationship between surface

temperatures and measured $\delta^{18}O$ ratios in ice cores can be deduced, while in Central Greenland this relation is weaker. These spatial differences might be explained by the fact that isotope changes are an integrated signal of the meso-scale variability

of atmospheric processes (Dansgaard, 1964; Merlivat and Jouzel, 1979; Gat, 1996), which might partially be decoupled from surface temperature changes in homogeneous terrain.

Consistent patterns over Greenland are also modelled for the interannual temporal $\delta^{18}O$-temperature slope in the mid-Holocene and the present-day simulation. But in comparison to the spatial $\delta^{18}O$-temperature slope, the interannual temporal interrelations between the surface temperature and $\delta^{18}O$ are rather small. This weaker interannual $\delta^{18}O$-temperature slope is

490 again in line with the results of Sjolte et al. (2011).

The presented study demonstrates that the isotope-enabled MPI-ESM-wiso - COSMO_iso model chain with realistically implemented stable water isotope fractionation processes constitutes a useful supplement to reconstruct regional paleo-climate conditions during the mid-Holocene in Greenland. By means of such an isotope-enabled GCM-RCM model chain, locally measured isotope ratios in an ice core can be adequately linked to spatially coarse climate model results and

495 conclusions on the underlying climatic processes leading to these ratios can be drawn in a physically consistent way. This approach might also be very helpful for other isotope-enabled GCMs and understanding their deviations from observed isotope ratios in different paleo-time periods and regions. Particularly in regions in which large differences occur between simulated and observed $\delta^{18}O$ ratios, due to small-scale orographic variations, like parts of Europe and North America (Cauquoin et al., 2019; Comas-Bru et al., 2019), an improved representation of small-scale processes can potentially reduce

these biases, and consequently, the reconstruction of regional paleo-climate patterns can become more reliable. To test this hypothesis, in follow-up studies, more time slices will be simulated with the presented MPI-ESM-wiso – COSMO_iso model chain for different periods and different regions.

*Code availability*. The isotope-enabled version COSMO_iso is available upon request from Marcus Breil. The code of the isotopic version MPI-ESM-wiso is available upon request on the AWI's GitLab repository (https://gitlab.awi.de/mwerner/mpi-esm-wiso, Cauquoin et al., 2019).

*Author contributions.* Marcus Breil performed the COSMO_iso mid-Holocene simulations, EC the COSMO_iso present-day simulation. The ECHAM5-wiso simulations were performed by Martin Werner, the MPI-ESM-wiso simulations by Alexandre Cauquoin. Melanie Karremann performed the comparison of the model results with present-day observations for the standard climatological parameters. Marcus Breil analysed the presented $\delta^{18}O$ model results and wrote the manuscript

with contributions from all co-authors.

*Competing interests.* The authors declare that they have no conflict of interest.

*Acknowledgements.* This work was supported by the German Federal Ministry of Education and Research (BMBF) as a Research for Sustainability initiative (FONA) through PalMod project (FKZ: 01LP1511B). All simulations were performed at the German Climate Computing Center (DKRZ). Marcus Breil acknowledges support from the KIT Publication Fund of the Karlsruhe Institute of Technology.

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
