# Peer review of "Applying an isotope-enabled regional climate model over the Greenland ice sheet: effect of spatial resolution on model bias"

_Climate of the Past, 2019_

## Referee Comment (RC1) · Anonymous Referee #1 · 9 Jun 2020

This manuscript presents first outputs of the COSMO-iso model for the Arctic regions over the present-day and mid-Holocene. The results are compared to measurements performed in snow and ice cores and the agreement is rather good, better than with a GCM, between model and data hence validating the use of a RCM equipped with isotopes to look at fine spatial scale the variability of water isotopes in this region.

Even if I am not very enthusiastic with this manuscript, this is a valuable contribution but I feel that the study could be developed a bit more following the comments given below. In general, I am a bit disappointed by the manuscript compared to the previous study on the same subject, Sjolte et al., 2011. This previous study using a regional model

with isotopes presented numerous applications especially on the temporal variability, an aspect which is fully absent here. Could perhaps the authors elaborate a bit more on the temporal variability (seasonal and interannual variability) and compare to available data or to this previous study ?

I understand that the authors like to focus their study on the mid-Holocene but it is not clear why. Also, the difference between mid-holocene and PST is not very large so that the comparison between the two periods is not the best to validate the temporal variability of the model. It is also complicated to perform such a comparison because COSMO-iso is associated with ECHAM-5 wiso for present-day and MPI-ESM-wiso for the mid Holocene. Without a comparison between ECHAM5-wiso and MPI-ESM-wiso which is not discussed here, it is quite complicated to perform comparison between mid-Holocene and PST. Was it really impossible to use the same GCM for both simulations ?

I am quite worried that the present study is submitted while the evaluation of the COSMO model (without isotopes) is not performed (cf sentences 66-67). Why then compared d18O values to observations if we have no validation of basic climatic parameters (temperature, etc...). At least some sentences for the most relevant parameters should be included here. I am quite surprised by the paragraph on fractionation at snow covered surfaces. For the work on the Arctic, you have a large number of paper co-authored by Hans Christian Steen Larsen which discuss the isotopic equilibrium or disequilibrium between surface snow, precipitation and water vapor in Greenland. It is quite strange to use a dataseries from Karlsruhe to calibrate fractionation between snow and water vapor in Greenland when data are available there.

Similarly, I am surprised that you do not have more observations gathered in part 2.2. Why only concentrating on core top while you have some series of observations (Bonne et al., ACP, 2014; papers co-authored by Steen-Larsen). You may also want to include the core studied by Furukawa et al., JGR, 2017).

I am not so convinced by figure 4b and the associated discussion stating that the bias are very small. First, the scale is much to large, it would be enough to draw the y-axis between -2 and +2 permil. And then, you obtain opposite variations between the red (model, negative d18O anomaly) and orange (observation, positive d18O anomaly) so that the comparison of the results is actually not convincing even if the changes are small in both cases but this is expected sine Mid Holocene is not very different from PI. I see this point as a strong weakness.

It would have been nice to discuss the temporal d18O vs Temperature gradient and not only the local spatial one. Also, we are awaiting some discussions / perspectives on the implications of these calculated spatial gradients for ice core interpretation. It would be nice to elaborate on this.

Other comments to consider: - I do not understand the following sentence in the abstract: "Furthermore, by investigating the $\delta$18O ratios in all COSMO_iso grid boxes located within the corresponding ECHAM5-wiso grid box, the observed isotopic ratios can be classified as a possible local $\delta$18O ratio within the spatial uncertainties, derived by the regional downscaling approach."

- This sentence in the abstract is not very concrete "But again, the range of the COSMO_iso_50km $\delta$18O variability in the corresponding MPI-ESM-wiso grid boxes around each station is consistent with the observed $\delta$18O values"

- I am surprised in the introduction by the discussion about mid-holocene. In Greenland, the temperature better seems on a plateau between the beginning of the Holocene (optimum) and the mid-Holocene.

- L. 46: why do you discuss the ability of a GCM to reproduce the regional changes – why not discuss better the (dis)ability of a GCM equipped with isotopes to reproduce the regional changes of water isotopic composition.

- Table 1: Please correct the date for the reference of Weissbach et al., 2016. . . ; also

[Figure]

give the units for d18O

- It is very difficult to compare data and measurements on figure 1

- How is the yearly mean d18O value calculated ? Is there any weighting by the pre-cipitation amount ? Could this effect be discussed when compared to the observations ?

- L. 289: I do not understand this sentence "At the coastline, the $\delta$18Otemperature

- gradient is low, reflecting the high surface temperature and $\delta$18O variability in this region" – in general the whole paragraph needs to be rewritten since it is largely unclear (last sentence of the paragraph is particularly vague -> to what mechanisms do you refer ?)

---

## Referee Comment (RC2) · Anonymous Referee #2 · 11 Jul 2020

**Review: The dependency of the $\delta^{18}O$ discrepancy between ice cores and model simulations on the spatial model resolution**

Marcus Breil, Emanuel Christner, Alexandre Cauquoin, Martin Werner, Gerd Schädler

This study examines outputs of a regional climate model (RCM) enabled to compute fractionation of water isotopes over the Greenland ice sheet. The COSMO_iso RCM is forced at the lateral boundaries with isotope enabled GCM simulations with atmospheric nudging. Outputs of COSMO_iso simulations for the present day and the mid-holocene (at a 50 km spatial resolution) are compared against ice core isotopic measurements. For the present-day simulations the RCM simulations generally improved the agreement with observations compared to the GCM results, with the improvements generally occurring in regions with coarser GCM resolution. Higher-resolution RCM simulations at 7 km did not further improve the agreement, producing a worse agreement in some instances. For the mid-Holocene simulations, there was not a large improvement resulting from the RCM simulations (although data were available only from four ice cores). The authors note that the higher-resolution simulations provide a range of spatial variability for the coarse resolution grid that can be used to generate a distribution for comparison against ice core measurements. They also examine gradients of isotope ratio relative to temperature, finding higher variability in temperature and isotope ratios along the ice sheet margins.

**General Comments**

In general, the study appears to be scientifically sound, and well-organized. The work represents an important step in developing an improved understanding of the relationship between measured isotopic ratios and historical climate. The presentation, particularly the language, needs improvement, with many grammatical errors. The figures are somewhat difficult to read at first glance and also require improvements. I also have some concerns about the manuscript, in particular:

1. The in situ measurements are all located within the high-elevation center of the ice sheet, with one exception. It is therefore difficult to evaluate the degree to which the model simulations capture the spatial variability. While the RCM simulation improves the agreement with the southern-most observations, it introduces a positive bias in the north. It seems this could be due to differences in the dynamical simulation in the RCM relative to the ESM rather than increased variability in the higher resolution RCM as the authors argue.

2. Given the above points, the added value of the RCM simulation is not entirely clear, even in the present-day simulation, although the plots seem to suggest that it does provide some improvement in the mean value. The authors should provide quantitative estimates as to the improvement associated with the RCM.

3. The method of averaging observational data (which may contain missing values) is not entirely clear. The authors have not discussed potential errors in the observations.
4. I think the authors' approach of using the high-resolution variability as an indicator of the potential spatial variability within a coarse resolution grid cell, that can then inform the point observation to model grid cell comparison, is interesting. If the authors can find any literature supporting this argument, I think this would strengthen the manuscript.
5. This is not essential but the presentation of the manuscript could be improved if the authors use a different projection that doesn't distort the Greenland ice sheet, and if they label figures with brief headings that summarize each sub-figure without necessitating a thorough reading of the caption.

**Specific Comments**

1. **Title:** The title could be improved to better describe the study. The title should include mention of Greenland and types of models that are used. Possible revision: "Applying an isotope-enabled regional climate model over the Greenland Ice Sheet: effect of spatial resolution on model bias"
2. **Lines 7-9:** The authors should mention here the motivation and purpose of the study, which is described well in the introduction section.
3. **Line 9:** Change "isotopic ratios in Greenland" to "isotopic ratios in Greenland ice cores".
4. **Line 10:** Explain that ECHAM5-wiso and MPI-ESM-wiso are GCM simulations and spell out acronyms.
5. **Lines 15-16:** This sentence is confusing. Suggest revising to something like: "…the COSMO_iso estimates provide a distribution of values representing spatial uncertainty that give context to comparison with observed isotopic ratios."
6. **Lines 20-23:** These sentences are confusing. I think the authors can simply say something like: "Despite the lack of improvement in model biases, the RCM simulations provide a distribution that allow the effects of spatial uncertainty to be taken into account in the comparison between point measurements and model outputs."
7. **Line 60:** The authors mention temporal resolution here, but this is not discussed in the rest of the manuscript. I suggest providing further details here about temporal downscaling and noting that the focus of the present study is on spatial downscaling.
8. **Lines 70-75:** The text here repeats some information that was mentioned earlier. Suggest revising to avoid repetition.
9. **Line 92:** It should be first noted here that snow surface albedo is fixed and is not spatially and temporally variable.
10. **Lines 120-144:** How are the ocean boundary conditions specified? Are these from reanalysis data?

11. **Line 111:** What is meant by "the models"? Please clarify.
12. **Lines 114-119:** Are the authors referring to work they have performed comparing COSMO_iso to observations, or is this referring to the Christner et al. (2017) study? Please clarify. Also, please clarify how the processes are treated in the COSMO_iso model.
13. **Lines 123-124:** Note the domain boundaries for the Arctic simulation.
14. **Lines 128-130:** Is this an additional simulation forced by the coarse resolution run, or a nested domain within the larger domain?
15. **Line 130:** What is meant by "technical reasons"? Please clarify.
16. **Lines 152-153:** How are the authors dealing with missing data? If there are large temporal gaps in some of the datasets this could influence the average values.
17. **Table 1:** Are all the datasets available for the specified period? What is the effect of missing data on the estimates? Does the depth of the cores/snow pits affect the average? Please comment and perhaps perform calculations to assess these affects.
18. **Line 183:** What is the average reduction in the bias?
19. **Lines 199-205:** I don't quite understand the logic here. I think what the authors are saying is that the high-resolution simulation leads to a higher degree of variability in locally simulated values. Due to the uncertainty in the model simulation, this may lead to a larger bias with respect to in situ point measurements, which may actually be closer to the average value on the coarse resolution grid. However, running the high resolution simulation allows for computation of a range of local variability, which can be used to compare model to observed values, accounting for the inherent uncertainty of the in situ measurement associated with local variability. This is an interesting and reasonable argument. I think the authors need to articulate it better here. Also if the authors can find any literature showing similar results this would be helpful in supporting this argument.
20. **Figure 2:** Why are sites 17 and 18 missing here? Are data from these locations missing for this year? Please clarify in the caption and in the main text.
21. **Lines 223-228:** This argument does not make sense to me. Looking at the box plots in Figure 3, the variability for these particular stations does not seem to be larger here than at other locations. Rather, there appears to simply be a model bias at this location. One can also see from Figure 1, that COSMO_iso seems to shift the low isotope values in central northern Greenland further south relative to the ECHAM5-wiso, thereby increasing the bias in these areas somewhat. The authors should clarify or revise their arguments here.
22. **Lines 251-257:** This paragraph would more appropriately follow the first paragraph of the section, detailing the mid-Holocene results.
23. **Figure 4:** The y-axis label is confusing. Suggest changing to $\delta^{18}O$ difference. In the caption labels, suggest replacing with MPI_ESM_wiso – obs. and COSMO_iso_50km – obs.

24. **Line 261:** Is the green point for the 50 km grid cell closest to the measurement location? Please clarify.
25. **Line 263:** Spell out PI.
26. **Lines 266 – 294:** I suggest making this a new section, discussing sub-ESM-grid variability.
27. **Line 286:** Calling this a temperature gradient suggest that it is a change in temperature with elevation. Is this indeed a gradient, established through a linear fit of isotope ratio vs. temperature for the sub-grid results for each grid cell, or is it simply a ratio of the standard deviation? Please clarify by revising the text here.
28. **Line 294:** Change "the same mechanisms" to "similar mechanisms".
29. **Figure 5:** Site 1 is very difficult to see here and in other figures. Is there a way to improve visibility, perhaps by changing colors? Also label the color axis "$\delta^{18}O$ standard deviation" and "temperature standard deviation [K]" for clarity.
30. **Line 301:** Change "Simulated variability" to "simulated sub-grid-scale variability".
31. **Figure 6:** This colormap is likely not suitable for red-green colorblind readers. Suggest using a different colormap.
32. **Lines 330-331:** As noted earlier, in some cases this may be a result of increased variability, but it could also be a bias introduced in the RCM simulation.
33. **Line 343:** Suggest changing "The same" to "Similar".
34. **Line 358:** Change "prove" to "test".

**Technical Corrections**

1. **Line 7:** spell out RCM at the beginning of the line: "isotope-enabled Regional Climate Model (RCM) for Greenland. The capability of the applied RCM COSMO_iso,…"
2. **Line 13:** Change "a downscaling" to "dynamical downscaling" for clarity.
3. **Lines 14-15:** Revise to "yields improvements only for coastal areas with complex terrain."
4. **Line 19:** Change "already on a high level" to "already agrees well with observations"
5. **Line 26:** Change "deviations to" to "deviations relative to"
6. **Line 32:** Change "like past changes of temperature, out of " to "such as past temperature changes using"
7. **Line 37:** Change "was steadily rising" to "steadily rose"
8. **Line 39:** Change "were steadily decreasing" to "steadily decreased".
9. **Line 40:** Change "took place" to "had taken place".
10. **Lines 41-42:** Suggest revising to read "period of particular interest, given recent Arctic warming, as it was characterized by Arctic warming resulting from orbital forcing…"
11. **Line 43:** Change "processes, leading to this warming," to "processes leading to this warming…"

12. **Line 44:** Suggest changing "reflect" to "reproduce".
13. **Line 46:** Remove "which are" before "documented in".
14. **Line 51:** Suggest changing "does not meet" to "does not reproduce" or "does not adequately represent"
15. **Line 54:** Change "also often not entirely resolved" to "not well resolved" and "coarsely resolved GCMs" to "coarse resolution GCMs"
16. **Line 56:** Change "deviations to" to "deviations relative to"
17.  **Lines 63-64:** Suggest changing to "investigated, and the impact of such small-scale spatial variability on the discrepancy between simulated and observed paleo-climate conditions in the Arctic region is examined.
18.  **Line 67:** Change "separated" to "separate".
19.  **Line 82:** Spell out "COSMO".
20.  **Line 87:** Change "presented" to "present".
21. **Line 100:** Change "2 m temperature" to "2 m air temperature" for clarity.
22. **Line 114:** Add "the" before "best agreement"
23. **Line 121:** Change "reflect" to "reproduce".
24. **Line 134:** Change "simulation has been" to "simulation is"
25. **Line 138:** Is the improvement to surface albedo for all surface types or one particular surface type?
26. **Line 147:** Perhaps remove "different" from before "different observational data".
27. **Line 151:** Remove "used" before "$\delta^{18}O$ values".
28. **Line 172:** Change "models capability" to "models' capability".
29. **Line 175:** Change "decline stronger" to "decline more rapidly".
30. **Line 179:** Change "stonger pronounced" to "more pronounced.
31. **Line 181:** Change "at which" to "for which".
32. **Line 182:** Change "deviations to" to "deviations from".
33. **Line 185:** Change "results anymore" to "results further".
34. **Line 188:** Change "a complex terrain" to "complex terrain"
35. **Line 194:** Change "a higher agreement" to "an improved agreement".
36. **Line 196:** Change "an enlarged heterogeneity" to "an increased heterogeneity".
37. **Line 236:** Change "differences for" to "differences between" and "grid box results to the" to "grid box results and the"
38. **Line 238:** Change "shown as Box-Whiskers" to "shown as a Box-Whiskers".
39. **Lines 277-278:** Change "the three regions…" to "in three regions of Greeenland with substantially different sub-pixel isotopic ratio variabilities."
40. **Line 281:** Change "exhibiting also regional variations" to "which also exhibits regional variations…"
41. **Line 283:** Change "does consequently not only depend" to "consequently not only depends"
42. **Line 313:** Change "agreement to climate" to "agreement with climate"
43. **Lines 322-324:** Revise to "But for northern Greenland, regional climate simulations with COSMO_iso increase the bias with respect to observations and

---

## Author Comment (AC1) · 18 Aug 2020

- Dear Reviewer. Thank you very much for your constructive comments. We think that you addressed some important issues and we hope that we are able to respond satisfactorily.

This manuscript presents first outputs of the COSMO-iso model for the Arctic regions over the present-day and mid-Holocene. The results are compared to measurements performed in snow and ice cores and the agreement is rather good, better than with a GCM, between model and data hence validating the use of a RCM equipped with isotopes to look at fine spatial scale the variability of water isotopes in this region.
Even if I am not very enthusiastic with this manuscript, this is a valuable contribution but I feel that the study could be developed a bit more following the comments given below. In general, I am a bit disappointed by the manuscript compared to the previous study on the same subject, Sjolte et al., 2011. This previous study using a regional model with isotopes presented numerous applications especially on the temporal variability, an aspect which is fully absent here. Could perhaps the authors elaborate a bit more on the temporal variability (seasonal and interannual variability) and compare to available data or to this previous study?
- Beside an increased spatial variability, RCMs can show a different (increased) temporal variability in comparison to GCMs. These differences in the temporal variability can, of course, lead to differences in the yearly mean values, as shown by Sjolte et al., (2011) for systematic δ18O biases in different seasons. In addition, such seasonal δ18O differences can be used to reveal systematic model deficiencies related to, for example, large-scale circulation patterns (Werner et al. 2000), in turn affecting the interpretation of paleo-climate periods.
In order to investigate this potential impact, an analysis of the temporal δ18O variability in precipitation in the present-day GCM and RCM results is added to the manuscript. In this context, the simulated monthly δ18O values are compared to observed monthly δ18O values in precipitation, collected at arctic stations of the Global Network of Isotopes in Precipitation (GNIP). In general, the modeled δ18O values in precipitation of COSMO-iso are in good agreement with the monthly GNIP data (Figure a, Figure 5 in the manuscript). But in contrast to Sjolte et al., (2011), no systematic over- or underestimation of observed isotope ratios is simulated with the RCM. This is true for each season. Neither in winter (low δ18O values), nor in summer (high δ18O values) systematic deviations to the observations are simulated. Thus, the seasonal variability in the COSMO-iso results has no systematic impact on the yearly mean δ18O values and is therefore not the reason for systematic differences between model results and observations.
In order to investigate the interannual variability in the simulation results, an analysis of the temporal δ18O-temperature slope is included in the manuscript, in addition to the spatial δ18O-temperature slope analysis (Figure b, included in Figure 7 and 9 in the manuscript). This temporal δ18O-temperature slope is calculated for both periods, present-day and mid-Holocene, based on the yearly mean isotope and temperature values. The results show that the temporal δ18O-temperature slope is in both periods smaller than the spatial slope, which is in accordance with the results of Sjolte et al., (2011). The interannual δ18O variations are consequently all over Greenland rather small and lowly correlated with the surface temperatures. The impact of temporal surface temperature variations on the temporal δ18O variability is therefore small in Greenland.

[Figure]

Figure a: Monthly δ18O simulated with COSMO_iso_50km for the period 2008 - 2014 and the corresponding observations for 9 GNIP stations

[Figure]

Figure b: Temporal δ18O-temperature slope for Greenland for the present-day (left) and the mid-Holocene (right)

I understand that the authors like to focus their study on the mid-Holocene but it is not clear why. Also, the difference between mid-holocene and PST is not very large so that the comparison between the two periods is not the best to validate the temporal variability of the model.
- We chose the mid-Holocene for our plaeo-climate simulations since it is a period of particular interest for Greenland. By that time an Arctic warming took place due to orbital forcing variations and their

related feedbacks on large-scale climate variations, which exhibits similarities to the strong recent Arctic warming. Thus, the mid-Holocene provides the opportunity to investigate the processes, leading to this warming, in more detail and to potentially obtain new insights about the future development of the Arctic region (Yoshimori and Suzuki, 2019). Reliable model data are therefore particularly important to consistently analyze the associated processes.

It is also complicated to perform such a comparison because COSMO-iso is associated with ECHAM-5 wiso for present-day and MPI-ESM-wiso for the mid Holocene. Without a comparison between ECHAM5-wiso and MPI-ESM-wiso which is not discussed here, it is quite complicated to perform comparison between mid-Holocene and PST. Was it really impossible to use the same GCM for both simulations?

- Unfortunately, no present-day MPI-ESM-wiso simulations with dynamical fields nudged to reanalyses exist, as for ECHAM5-wiso. But we wanted such a nudged present-day reference simulation to assess the COSMO-iso model under the best possible conditions.

I am quite worried that the present study is submitted while the evaluation of the COSMO model (without isotopes) is not performed (cf sentences 66-67). Why then compared d18O values to observations if we have no validation of basic climatic parameters (temperature, etc...). At least some sentences for the most relevant parameters should be included here.

- the short discussion of the general model performance of COSMO in Greenland, regarding the standard climatic parameters in present-day simulations, is extended in the manuscript (see the new section 3.1.1 which is about the assessment of standard climatological parameters). For this purpose, a new figure about the differences between the simulated 2 m temperatures and precipitation sums to the observed ones, is now included (Figure c, Figure 2 in the manuscript). For this validation, observed temperatures and precipitation amounts in Greenland, collected by the Danish Meteorological Institute, are used (the locations of these stations are listed in Table 1 in the revised manuscript).

Both, simulated 2 m temperature as well as precipitation sums are in good agreement with the observations. Thus, the model is generally able to simulate reasonable near-surface temperatures and precipitation amounts for Greenland and can therefore be used for isotope applications in this region. A detailed analysis of the COSMO performance in Greenland is presented in Karremann et al., (2020).

[Figure]

Figure c: Simulated yearly mean (a) 2 m temperatures and (b) precipitation sums of a standard COSMO simulation, driven with ERA-Interim, for Greenland over the period 1995-2015 compared to DMI observations.

I am quite surprised by the paragraph on fractionation at snow covered surfaces. For the work on the Arctic, you have a large number of paper co-authored by Hans Christian Steen Larsen which discuss the isotopic equilibrium or disequilibrium between surface snow, precipitation and water vapor in Greenland. It is quite strange to use a dataseries from Karlsruhe to calibrate fractionation between snow and water vapor in Greenland when data are available there.

- The phrasing of this paragraph was misleading. We did not calibrate the fractionation during sublimation at snow covered surfaces. An equilibrium fractionation was assumed for surface layer snow and sea ice. Simulation results with this approximation were just additionally compared to an observational dataset in Karlsruhe. To avoid confusion, the paragraph is rephrased (Lines 119-122):

"To approximate this complex interplay of different influencing factors, in this study, an equilibrium fractionation during sublimation from surface layer snow and sea ice is assumed. However, the authors are aware that this is just a simplified description of isotope fractionation during sublimation."

Similarly, I am surprised that you do not have more observations gathered in part 2.2.Why only concentrating on core top while you have some series of observations (Bonne et al., ACP, 2014; papers co-authored by Steen-Larsen). You may also want to include the core studied by Furukawa et al., JGR, 2017).

- Thanks for the indication on further observational data sets. In the revised paper we included the data set of Furukawa et al., (2017) in our analysis (see Figures 3, 4 and 6) as you suggested. Additionally, we included $\delta18O$ data at GNIP stations in the manuscript (see Figure a and Figure 5 in the revised paper) to analyze the temporal variability of the simulated $\delta18O$ values in precipitation in COSMO-iso.

I am not so convinced by figure 4b and the associated discussion stating that the bias are very small. First, the scale is much to large, it would be enough to draw the y-axis between -2 and +2 permil. And then, you obtain opposite variations between the red (model, negative d18O anomaly) and orange (observation, positive d18O anomaly) so that the comparison of the results is actually not convincing even if the changes are small in both cases but this is expected since Mid Holocene is not very different from PI. I see this point as a strong weakness.

- For this figure (now Figure 8b), the same y-axis scale was used as in Figure 4a (now Figure 8a) to keep the results comparable. You are right, the sign is different for 6ka-PI_MPI and 6ka-OBS. But this is due to the small deviations between 6ka-PI_MPI and 6ka-OBS. Already small differences can consequently result in a changing sign. The message of this figure is that the high agreement of MPI-ESM-wiso in the mid-Holocene is not achieved by chance. This can be demonstrated by the small deviations between 6ka-PI_MPI and 6ka-OBS, even if these small differences show a different sign.

It would have been nice to discuss the temporal d18O vs Temperature gradient and not only the local spatial one.

- An analysis of the temporal $\delta18O$-temperature slope is now included in the manuscript. See comment above and Figure b.

Also, we are awaiting some discussions / perspectives on the implications of these calculated spatial gradients for ice core interpretation. It would be nice to elaborate on this.

- the results of this study show that a bias in GCM results does not inevitably contradict the measured isotope ratios in an ice core. The measured isotope ratios are potentially included, but hidden within the subgrid-scale uncertainty of a GCM grid box. Thus, a regional downscaling of GCM data is recommended. In this way, locally measured isotope ratios in an ice core can be adequately linked to spatially coarse climate model results and conclusions on the underlying climatic processes leading to

these ratios can be drawn in a physically consistent way. This point is now stronger emphasized in the discussion (Lines 432-444 and 449-453):

"As $\delta18O$ ratios are used as an indicator for temperatures in past climates (Dansgaard et al., 1969; Masson-Delmotte et al., 2005; Jouzel, 2013), it is important to understand how the presented COSMO_iso simulations might be able to improve these isotope-based temperatures reconstructions. In general, the regional surface temperature variability and the regional $\delta18O$ variability show similar patterns for Greenland. In both cases the variability is high at the coast and low on the inland plateau. Similar patterns as in the mid-Holocene can also be seen for the present-day simulations. These spatial variability patterns of $\delta18O$ and the surface temperature are in line with the results of Sjolte et al. (2011) for RCM simulations under present-day conditions for Greenland. Based on these variability patterns, it can be derived that the regional surface temperature variability highly depends on the surface characteristics in Greenland. However, for the regional isotopic ratio variability, this dependence appears to be less pronounced. At the coastline, a clear relationship between surface temperatures and measured $\delta18O$ ratios in ice cores can be deduced, while in Central Greenland this relation is weaker. These spatial differences might be explained by the fact that isotope changes are an integrated signal of the meso-scale variability of atmospheric processes (Dansgaard, 1964; Merlivat and Jouzel, 1979; Gat, 1996), which might partially be decoupled from surface temperature changes in homogeneous terrain."

"The presented study demonstrates that the isotope-enabled MPI-ESM-wiso - COSMO_iso model chain with realistically implemented stable water isotope fractionation processes constitutes a useful supplement to reconstruct regional paleo-climate conditions during the mid-Holocene in Greenland. By means of such an isotope-enabled GCM-RCM model chain, locally measured isotope ratios in an ice core can be adequately linked to spatially coarse climate model results and conclusions on the underlying climatic processes leading to these ratios can be drawn in a physically consistent way."

Other comments to consider:
- I do not understand the following sentence in the abstract: "Furthermore, by investigating the $\delta18O$ ratios in all COSMO_iso grid boxes located within the corresponding ECHAM5-wiso grid box, the observed isotopic ratios can be classified as a possible local $\delta18O$ ratio within the spatial uncertainties, derived by the regional downscaling approach."
This sentence in the abstract is not very concrete "But again, the range of the COSMO_iso_50km $\delta18O$ variability in the corresponding MPI-ESM-wiso grid boxes around each station is consistent with the observed $\delta18O$ values"
- both statements are rephrased in the revised manuscript (Lines (21-26):

"Despite this lack of improvements in model biases, the study shows that in both periods, observed $\delta18O$ values at measurement sites constitute isotope ratios which are mainly within the subgrid-scale variability of the global ECHAM5-wiso and MPI-ESM-wiso simulation results. The correct $\delta18O$ ratios are consequently already included but hidden in the GCM simulation results, which just need to be extracted by a refinement with an RCM. In this context, the RCM simulations provide a spatial $\delta18O$ distribution by which the effects of local uncertainties can be taken into account in the comparison between point measurements and model outputs."

I am surprised in the introduction by the discussion about mid-holocene. In Greenland, the temperature better seems on a plateau between the beginning of the Holocene (optimum) and the mid-Holocene.
- the text is adapted (Line 41-42):

"Between the early Holocene and the Holocene Thermal Maximum in the mid-Holocene (6 ka), a pronounced warm phase took place"

L. 46: why do you discuss the ability of a GCM to reproduce the regional changes –why not discuss better the (dis)ability of a GCM equipped with isotopes to reproduce the regional changes of water isotopic composition.
- we included a discussion about the disability of isotope-enabled GCMs to reproduce regional changes and the added value of isotope-enabled RCMs in the manuscript according to your suggestions (Lines 57-65):

"For stable water isotopes, key physical processes of isotope fractionation are therefore not well resolved in coarse resolution GCMs, leading to differences between simulated and observational isotope data, especially in complex terrains (Sturm et al., 2005; Werner et al., 2011). Isotope-enabled GCMs are consequently not able to reproduce regional changes in isotope ratios quantitively (e.g. Risi et al., 2010), and the simulated isotope ratios with GCMs exhibit in many cases larger deviations relative to observed ratios than the results of corresponding Regional Climate Model (RCM) simulations. For instance, Sturm et al., (2007) were able to reduce the bias of simulated isotope ratios in precipitation, by a regional downscaling of an isotope-enabled GCM run in South America. Comparable results were achieved by Sjolte et al., (2011) for isotope-enabled RCM simulations in Greenland."

Table 1: Please correct the date for the reference of Weissbach et al., 2016...; also give the units for d18O
- is corrected.

It is very difficult to compare data and measurements on figure 1
- Since we are aware of this, differences between simulated $\delta18O$ values and observed ones are additionally shown in Figure 2 (now Figure 4) as a bar plot.

How is the yearly mean d18O value calculated? Is there any weighting by the precipitation amount? Could this effect be discussed when compared to the observations?
- The modeled $\delta18O$ in precipitation is weighted with accumulation rate, i.e. months with high precipitation amounts get a higher weight compared to months with small precipitation amounts. We forgot to mention this in the manuscript. This is statement is now included (Lines 174-177).

L. 289: I do not understand this sentence "At the coastline, the $\delta18O$ temperature- gradient is low, reflecting the high surface temperature and $\delta18O$ variability in this region" – in general the whole paragraph needs to be rewritten since it is largely unclear (last sentence of the paragraph is particularly vague -> to what mechanisms do you refer?)
- the whole paragraph is rewritten and restructured (see Lines 385 – 395):

"The spatial surface temperature variability in the COSMO_iso_50km mid-Holocene simulation is shown in Figure 9b. Again, the mid-Holocene simulation shows the same surface temperature variability characteristics over Greenland as the present-day run with a high spatial variability near the coastline and almost no variability in Central Greenland. As a consequence, similar patterns of the spatial $\delta18O$-temperature slope are simulated for the mid-Holocene and the present-day, with low gradients at the coastline and high gradients in Central Greenland (Figure 9c). But in the mid-Holocene simulation, the contrast between the coastal regions and the inland plateau is less clearly pronounced than in the present-day run, due to the higher spatial $\delta18O$ variability in Central Greenland.

Nevertheless, the spatial δ18O-temperature interrelations are in both periods comparable. This is also the case for the temporal variabilities of δ18O and the surface temperature. As shown in Figure 9d, the annual δ18O-temperature slope is again very small over Greenland, although in some regions higher temporal slopes are simulated. But in principle, the influence of surface temperature variations on the temporal δ18O variability in the mid-Holocene is also small."

---

## Author Comment (AC2) · 18 Aug 2020

Review: The dependency of the d18O discrepancy between ice cores and model simulations on the spatial model resolution

Marcus Breil, Emanuel Christner, Alexandre Cauquoin, Martin Werner, Gerd Schädler

This study examines outputs of a regional climate model (RCM) enabled to compute fractionation of water isotopes over the Greenland ice sheet. The COSMO_iso RCM is forced at the lateral boundaries with isotope enabled GCM simulations with atmospheric nudging. Outputs of COSMO_iso simulations for the present day and the mid-holocene (at a 50 km spatial resolution)are compared against ice core isotopic measurements. For the present-day simulations the RCM simulations generally improved the agreement with observations compared to the GCM results, with the improvements generally occurring in regions with coarser GCM resolution. Higher-resolution RCM simulations at 7 km did not further improve the agreement, producing a worse agreement in some instances. For the mid-Holocene simulations, there was not a large improvement resulting from the RCM simulations (although data were available only from four ice cores). The authors note that the higher-resolution simulations provide a range of spatial variability for the coarse resolution grid that can be used to generate a distribution for comparison against ice core measurements. They also examine gradients of isotope ratio relative to temperature, finding higher variability in temperature and isotope ratios along the ice sheet margins.

**General Comments**
In general, the study appears to be scientifically sound, and well-organized. The work represents an important step in developing an improved understanding of the relationship between measured isotopic ratios and historical climate. The presentation, particularly the language, needs improvement, with many grammatical errors. The figures are somewhat difficult to read at first glance and also require improvements.
- Dear Reviewer. Thank you very much for your constructive and very detailed comments. We think that you addressed some important issues and we hope that we are able to respond satisfactorily to your comments.

I also have some concerns about the manuscript, in particular:
1. The in situ measurements are all located within the high-elevation center of the ice sheet, with one exception. It is therefore difficult to evaluate the degree to which the model simulations capture the spatial variability. While the RCM simulation improves the agreement with the southern-most observations, it introduces a positive bias in the north. It seems this could be due to differences in the dynamical simulation in the RCM relative to the ESM rather than increased variability in the higher resolution RCM as the authors argue.
- we agree and changed the argumentation according to your suggestion. Thank you very much for this helpful comment (Lines 231 – 235):

"As visible in Figure 3, these systematic differences are rather caused by a southward shift of the area of low yearly mean δ18O values in central Northern Greenland in COSMO_iso_50km relative to ECHAM5-wiso. As a result, the simulated δ18O values in central Northern Greenland in COSMO_iso_50km are higher than in ECHAM5-wiso. Since there, ECHAM5-wiso has already a high agreement with the observed δ18O values, a model bias is introduced in COSMO_iso_50km, causing the deviations relative to the observations in Northern Greenland."

2. Given the above points, the added value of the RCM simulation is not entirely clear, even in the present-day simulation, although the plots seem to suggest that it does provide some improvement in

the mean value. The authors should provide quantitative estimates as to the improvement associated with the RCM.
- we agree and mention now quantitative estimates of the RCM improvements in the present-day simulations (Lines 217 and 246). The average bias reduction of COSMO_iso_50km over all snow pit samples is 0.7‰, the average reduction of COSMO_iso_7km is 0.6‰.

3. The method of averaging observational data (which may contain missing values) is not entirely clear. The authors have not discussed potential errors in the observations.
- of course, observations are also associated with uncertainties. The impacts of firn diffusion, post-depositional erosion of surface snow by wind and the spatial uncertainties related to micrometeorological effects are now discussed in the revised paper (e.g. Lines 167-173). Since the used observational datasets do not contain missing data, no special averaging method is applied (See comments 16 & 17).

"Since all snow pit samples cover different time periods, the present-day $\delta18O$ values (black numbers in Table 2) are calculated as an average of all available $\delta18O$ values measured between 1940 and 2007. With this procedure uncertainties in snow pit samples and top ice core samples, associated with post depositional diffusion and the resulting constraints in analysing annual and interannual top ice core data (e.g. Johnson et al., 2000), can be neglected. However, further uncertainties in snow pit samples and ice core data remain, regarding the timescale assignment (Steig et al., 2005) and the spatial variability (Weißbach et al., 2016b)."

4. I think the authors' approach of using the high-resolution variability as an indicator of the potential spatial variability within a coarse resolution grid cell, that can then inform the point observation to model grid cell comparison, is interesting. If the authors can find any literature supporting this argument, I think this would strengthen the manuscript.
- With the publication of Shi et al., (2020), an additional reference substantiating our argumentation is now cited in the revised manuscript. In this study, the importance of small-scale processes to understand the measured water isotope variability is highlighted. According to this, GCM deficiencies in simulating this isotope variability are therefore caused by the missing representation of such small-scale processes in GCM simulations (this statement is no included in Lines 429-431). However, the authors are not aware of any further supporting literature.

Shi et al., (2020): https://agupubs.onlinelibrary.wiley.com/doi/full/10.1029/2019JD031751

5. This is not essential but the presentation of the manuscript could be improved if the authors use a different projection that doesn't distort the Greenland ice sheet, and if they label figures with brief headings that summarize each sub-figure without necessitating a thorough reading of the caption.
- as you already mentioned in your first comment, most of the point measurements are located in Central and Northern Greenland very close to each other. The chosen projection is therefore beneficial to better distinguish between these observations (especially in Figure 3). For that reason, we would like to stick to this projection.
In the revised manuscript, all figures are labeled with headings.

**Specific Comments**

1. Title: The title could be improved to better describe the study. The title should include mention of Greenland and types of models that are used. Possible revision: "Applying an isotope-enabled regional climate model over the Greenland Ice Sheet: effect of spatial resolution on model bias"
- the title is changed as you suggested. Thank you very much for your suggestion

2. Lines 7-9: The authors should mention here the motivation and purpose of the study, which is described well in the introduction section.
- the motivation of the study is now mentioned at the beginning of the abstract (lines 8-10):

"In order to investigate the impact of spatial resolution on the discrepancy between simulated δ18O and observed δ18O in Greenland ice cores, regional climate simulations are performed with the isotope-enabled Regional Climate Model (RCM) COSMO_iso."

3. Line 9: Change "isotopic ratios in Greenland" to "isotopic ratios in Greenland ice cores".
- is changed.

4. Line 10: Explain that ECHAM5-wiso and MPI-ESM-wiso are GCM simulations and spell out acronyms.
- in the revised manuscript, it is now mentioned that ECHAM5-wiso and MPI-ESM-wiso are isotope-enabled GCMs. The acronym MPI-ESM-wiso is now spelled out in the model description section 2.1.2. Since the GCM ECHAM is well-known in the modelling community and its acronym is very complex (a combination of **EC**MWF, which is already an acronym, and **Ham**burg, the location of the Max-Planck-Institute), we decided to not spell out ECHAM.

5. Lines 15-16: This sentence is confusing. Suggest revising to something like: "...the COSMO_iso estimates provide a distribution of values representing spatial uncertainty that give context to comparison with observed isotopic ratios."
- the abstract is rephrased in consideration of your suggestions (see comment 6).

6. Lines 20-23: These sentences are confusing. I think the authors can simply say something like: "Despite the lack of improvement in model biases, the RCM simulations provide a distribution that allow the effects of spatial uncertainty to be taken into account in the comparison between point measurements and model outputs."
- the abstract is rephrased in consideration of your suggestions (Lines 21-26):

"Despite this lack of improvements in model biases, the study shows that in both periods, observed δ18O values at measurement sites constitute isotope ratios which are mainly within the subgrid-scale variability of the global ECHAM5-wiso and MPI-ESM-wiso simulation results. The correct δ18O ratios are consequently already included but hidden in the GCM simulation results, which just need to be extracted by a refinement with an RCM. In this context, the RCM simulations provide a spatial δ18O distribution by which the effects of local uncertainties can be taken into account in the comparison between point measurements and model outputs."

7. Line 60: The authors mention temporal resolution here, but this is not discussed in the rest of the manuscript. I suggest providing further details here about temporal downscaling and noting that the focus of the present study is on spatial downscaling.
- in the revised manuscript, the text is adjusted as follows (Lines 66-71):

"Therefore, in the presented study, isotope-enabled GCM simulation results for the Arctic region are dynamically downscaled with an isotope-enabled RCM to a higher temporal and spatial resolution. By means of such regional simulations, the spatial and temporal variability of the isotopic ratios in the Arctic is potentially increased, accounting for the heterogeneity of local conditions at the different ice core locations and the associated uncertainties. In this way, the impact of highly resolved local conditions on the spatial and temporal variability of isotopic ratios is investigated, and the impact of such small-scale variability on the discrepancy between simulated and observed paleo-climate conditions in the Arctic region is examined."

According to this, an analysis of the temporal variability is additionally included in the paper (Figure 7d and 9d).

8. Lines 70-75: The text here repeats some information that was mentioned earlier. Suggest revising to avoid repetition.
- this information was only mentioned in the abstract. Therefore we would like to keep it in the text.

9. Line 92: It should be first noted here that snow surface albedo is fixed and is not spatially and temporally variable.
- snow surface albedo is not fixed. An alteration of the snow albedo with growing age is considered in the model. The increase in the snow albedo value from 0.7 to 0.8 refers to the albedo value of fresh snow. This is now specified in the manuscript (Line 97).

10. Lines 120-144: How are the ocean boundary conditions specified? Are these from reanalysis data?
- in the ECHAM5-wiso simulations, sea surface temperatures and sea ice cover are varying monthly based on ERA data. In MPI-ESM-wiso, the ocean component is calculated dynamically with the ocean model MPIOM. This is now mentioned in the text (Lines 131-132 and 151-152).

11. Line 111: What is meant by "the models"? Please clarify.
- we mean state-of-the-art isotope-enabled models. This is now clarified.

12. Lines 114-119: Are the authors referring to work they have performed comparing COSMO_iso to observations, or is this referring to the Christner et al. (2017) study? Please clarify. Also, please clarify how the processes are treated in the COSMO_iso model.
- these processes are not yet included in state-of-the-art isotope-enabled models. This is, for instance, discussed in Christner et al., (2017). The paragraph is rephrased to avoid confusion and to clarify how fractionation at snow covered surfaces is treated in COSMO_iso (Lines 114-122):

"Isotope fractionation during sublimation from a surface snow layer is poorly understood. Several different processes are suggested to be involved, which are not yet taken into account in state-of-the-art isotope enabled models (see e.g. discussion in Christner et al., 2017), such as non-fractionating layer-by-layer sublimation (e.g. Ambach et al. 1968), kinetic fractionation during sublimation into sub-saturated air, a diurnal cycle of sublimation combined with fractionating vapor deposition on the snow (e.g. Steen-Larsen et al., 2014), and fractionating melt water evaporation combined with recrystallization of residual melt water have been suggested (Gurney and Lawrence, 2004). To approximate this complex interplay of different influencing factors, in this study, an equilibrium fractionation during sublimation from surface layer snow and sea ice is assumed. However, the authors are aware that this is just a simplified description of isotope fractionation during sublimation."

13. Lines 123-124: Note the domain boundaries for the Arctic simulation.
- an additional figure showing the model domains (50 km and 7 km) is now included in the manuscript (Figure 1)

14. Lines 128-130: Is this an additional simulation forced by the coarse resolution run, or a nested domain within the larger domain?
- this simulation is nested in the 50 km simulation with COSMO_iso. This is now clarified in the text (Line 134-135).

15. Line 130: What is meant by "technical reasons"? Please clarify.
- this statement is removed from the text.

16. Lines 152-153: How are the authors dealing with missing data? If there are large temporal gaps in some of the datasets this could influence the average values.
- In the selected time periods, no missing data occurred in the yearly d18O values of the snow pit samples.

17. Table 1: Are all the datasets available for the specified period? What is the effect of missing data on the estimates? Does the depth of the cores/snow pits affect the average? Please comment and perhaps perform calculations to assess these affects.
- No, not all samples cover the whole period. But the individual datasets are consistent in themselves and do not contain missing data. In addition, the averaging periods of the respective snow pit samples are long enough to rule out statistical outliers.

18. Line 183: What is the average reduction in the bias?
- The average bias reduction of COSMO_iso_50km is 0.7‰, the average reduction of COSMO_iso_7km is 0.6‰. This is now mentioned in the text (Lines 217 and 246).

19. Lines 199-205: I don't quite understand the logic here. I think what the authors are saying is that the high-resolution simulation leads to a higher degree of variability in locally simulated values. Due to the uncertainty in the model simulation, this may lead to a larger bias with respect to in situ point measurements, which may actually be closer to the average value on the coarse resolution grid. However, running the high resolution simulation allows for computation of a range of local variability, which can be used to compare model to observed values, accounting for the inherent uncertainty of the in situ measurement associated with local variability. This is an interesting and reasonable argument. I think the authors need to articulate it better here. Also if the authors can find any literature showing similar results this would be helpful in supporting this argument.
- we rephrased the paragraph (Lines 264-267 and 271-276). Thank you very much for your helpful suggestions.

"As a consequence, an additional spatial variability is introduced in the RCM simulations in comparison to the GCM results. Due to uncertainties accompanied by model simulations, this can potentially increase the RCM bias with respect to in situ point measurements, which may actually be closer to the spatially averaged values simulated by the coarse GCM model."

"However, by performing higher resolved RCM simulations, the subgrid-scale variability of δ18O within GCM grid boxes can be simulated and compared to observed δ18O values. In this way, the inherent uncertainty of in situ measurements, associated with a local micrometeorological variability, can be considered. Thus, in the following, snow pit samples are not anymore solely compared to the

model grid boxes covering the samples location. Instead, it is investigated whether the δ18O range of all adjacent RCM grid boxes to a snow pit location is consistent with the observed δ18O value of the same site. For this, all RCM grid boxes located within the corresponding GCM grid box are included in the comparison with the observations."

20. Figure 2: Why are sites 17 and 18 missing here? Are data from these locations missing for this year? Please clarify in the caption and in the main text.
- this is corrected in the revised manuscript. Now the corresponding figures show all data points (now 19).

21. Lines 223-228: This argument does not make sense to me. Looking at the box plots in Figure 3, the variability for these particular stations does not seem to be larger here than at other locations. Rather, there appears to simply be a model bias at this location. One can also see from Figure 1, that COSMO_iso seems to shift the low isotope values in central northern Greenland further south relative to the ECHAM5-wiso, thereby increasing the bias in these areas somewhat. The authors should clarify or revise their arguments here.
- We agree with you and adapted our argumentation according to your suggestions (see general comment 1). Thank you very much for this helpful comment.

22. Lines 251-257: This paragraph would more appropriately follow the first paragraph of the section, detailing the mid-Holocene results.
- this paragraph is relocated according to your suggestions.

23. Figure 4: The y-axis label is confusing. Suggest changing to d18O difference. In the caption labels, suggest replacing with MPI_ESM_wiso –obs. and COSMO_iso_50km –obs.
- We changed the labeling of Figure 4 (now Figure 8) according to your suggestions.

24. Line 261: Is the green point for the 50 km grid cell closest to the measurement location? Please clarify.
- Yes it is. This is now clarified (Line 364).

25. Line 263: Spell out PI.
- is corrected.

26. Lines 266–294: I suggest making this a new section, discussing sub-ESM-grid variability.
- sections are new arranged in the revised manuscript according to your suggestions. Now, we discuss for both, present-day and mid-Holocene, first the simulated δ18O data in comparison to the point measurements and then the GCM δ18O subgrid-scale variability in sub-sections, respectively.

27. Line 286: Calling this a temperature gradient suggest that it is a change in temperature with elevation. Is this indeed a gradient, established through a linear fit of isotope ratio vs. temperature for the sub-grid results for each grid cell, or is it simply a ratio of the standard deviation? Please clarify by revising the text here.
- It is an isotope-temperature slope which constitutes a linear fit between the simulated δ18O ratios and the surface temperatures at all COSMO_iso_50km grid boxes within the respective GCM grid box. The isotope-temperature slope is a measure that is frequently used to analyze how strong isotope ratios and surface temperatures are interrelated. This is now clarified in the text (Lines 303-304):

"The spatial isotope-temperature slope constitutes a linear fit between the simulated δ18O ratios and the surface temperatures at all COSMO_iso_50km grid boxes within the respective ECHAM5-wiso grid box."

28. Line 294: Change "the same mechanisms" to "similar mechanisms".
- the sentence is rephrased in the revised manuscript.

29. Figure 5: Site 1 is very difficult to see here and in other figures. Is there a way to improve visibility, perhaps by changing colors? Also label the color axis "d18O standard deviation" and "temperature standard deviation[K]" for clarity.
- We changed the color of the markers to green in the corresponding figures. The labeling is changed according to your suggestions.

30. Line 301: Change "Simulated variability" to "simulated sub-grid-scale variability".
- is adapted in the revised manuscript.

31. Figure 6: This color map is likely not suitable for red-green colorblind readers. Suggest using a different color map.
- we changed the color map to blue-red.

32. Lines 330-331: As noted earlier, in some cases this may be a result of increased variability, but it could also be a bias introduced in the RCM simulation.
- This was actually a statement meant about the Renland station. This is corrected in the revised paper. Sorry for this mistake (Line 418-420):

"This in turn can lead to additional noise and thus, a deviating RCM behaviour with even an increase in the absolute model bias, as seen for the Renland station."

33. Line 343: Suggest changing "The same" to "Similar".
- is corrected

34. Line 358: Change "prove" to "test".
- is corrected

**Technical Corrections**

1. Line 7: spell out RCM at the beginning of the line: "isotope-enabled Regional Climate Model (RCM) for Greenland. The capability of the applied RCM COSMO_iso,..."
- is corrected.

2. Line 13: Change "a downscaling" to "dynamical downscaling" for clarity.
- is corrected.

3. Lines 14-15: Revise to "yields improvements only for coastal areas with complex terrain."
- is corrected.

4. Line 19: Change "already on a high level" to "already agrees well with observations"
- is corrected.

5. Line 26: Change "deviations to" to "deviations relative to"
- is corrected.

6. Line 32: Change "like past changes of temperature, out of" to "such as past temperature changes using"
- is corrected.

7. Line 37: Change "was steadily rising" to "steadily rose"
- is corrected.

8. Line 39: Change "were steadily decreasing" to "steadily decreased".
- is corrected.

9. Line 40: Change "took place" to "had taken place".
- is corrected.

10. Lines 41-42: Suggest revising to read "period of particular interest, given recent Arctic warming, as it was characterized by Arctic warming resulting from orbital forcing..."
- we keep the current phrasing

11. Line 43: Change "processes, leading to this warming," to "processes leading to this warming..."
- is corrected.

12. Line 44: Suggest changing "reflect" to "reproduce".
- is corrected.

13. Line 46: Remove "which are" before "documented in".
- is corrected.

14. Line 51: Suggest changing "does not meet" to "does not reproduce" or "does not adequately represent"
- is corrected.

15. Line 54: Change "also often not entirely resolved" to "not well resolved" and "coarsely resolved GCMs" to "coarse resolution GCMs"
- is corrected.

16. Line 56: Change "deviations to" to "deviations relative to"
- is corrected.

17. Lines 63-64: Suggest changing to "investigated, and the impact of such small-scale spatial variability on the discrepancy between simulated and observed paleo-climate conditions in the Arctic region is examined.
- is corrected.

18. Line 67: Change "separated" to "separate".
- this sentence is removed in the revised manuscript.

19. Line 82: Spell out "COSMO".

- is corrected.

20. Line 87: Change "presented" to "present".
- is corrected

21. Line 100: Change "2 m temperature" to "2 m air temperature" for clarity.
- is corrected

22. Line 114: Add "the" before "best agreement"
- this sentence is removed in the revised manuscript.

23. Line 121: Change "reflect" to "reproduce".
- is corrected

24. Line 134: Change "simulation has been" to "simulation is"
- we keep the current phrasing

25. Line 138: Is the improvement to surface albedo for all surface types or one particular surface type?
- For ECHAM6, a new land-albedo has been developed (Brovkin et al., 2013, JAMES, https://doi.org/10.1029/2012MS000169). But since we are focusing on Greenland in this study, different surface types are not so relevant. However, the albedo over sea ice area is also considered (treatment of melt ponds on sea ice). For land ice surface, the snow age is taken into account.

26. Line 147: Perhaps remove "different" from before "different observational data".
- is corrected

27. Line 151: Remove "used" before "d18O values".
- is corrected

28. Line 172: Change "models capability" to "models'capability".
- is corrected

29. Line 175: Change "decline stronger" to "decline more rapidly".
- is corrected

30. Line 179: Change "stonger pronounced" to "more pronounced.
- is corrected

31. Line 181: Change "at which" to "for which".
- is corrected

32. Line 182: Change "deviations to" to "deviations from".
- is corrected

33. Line 185: Change "results anymore" to "results further".
- is corrected

34. Line 188: Change "a complex terrain" to "complex terrain"
- is corrected

35. Line 194: Change "a higher agreement" to "an improved agreement".
- is corrected

36. Line 196: Change "an enlarged heterogeneity" to "an increased heterogeneity".
- is corrected

37. Line 236: Change "differences for" to "differences between" and "grid box results to the" to "grid box results and the"
- the sentence is rephrased in the revised manuscript.

38. Line 238: Change "shown as Box-Whiskers" to "shown as a Box-Whiskers".
- is corrected

39. Lines 277-278: Change "the three regions…"to "in three regions of Greenland with substantially different sub-pixel isotopic ratio variabilities."
- is corrected

40. Line 281: Change "exhibiting also regional variations" to "which also exhibits regional variations…"
- the sentence is rephrased in the revised manuscript.

41. Line 283: Change "does consequently not only depend" to "consequently not only depends"
- is corrected

42. Line 313: Change "agreement to climate" to "agreement with climate"
- is corrected

43. Lines 322-324: Revise to "But for northern Greenland, regional climate simulations with COSMO_iso increase the bias with respect to observations and
- is corrected

---

## Referee Report (RR1)

**Review:  The dependency of the $\delta^{18}O$ discrepancy between ice cores and model simulations on the spatial model resolution**

Marcus Breil, Emanuel Christner, Alexandre Cauquoin, Martin Werner, Gerd Schädler

**General Comments**
I feel the authors have addressed some of the concerns from both reviewers, but I still have some concerns about the manuscript.  In particular, although some additional quantitative evidence is now provided, I feel the authors have not supported their claims sufficiently with quantitative evidence, and additional quantitative analysis should be performed before the manuscript is published.   If the authors find that the analysis does not support their claims, they should modify their conclusions to reflect this analysis.  I think these changes will mostly be minor but since they have the potential to change the conclusions of the study I have selected major revisions.  Some remaining general comments are:

In **Section 3.1.2** the authors have computed the average reduction in bias for the RCM simulation.  But it is still unclear how much of an improvement this represents.  Is this a statistically significant difference?  Has the root mean squared error also been reduced in the RCM simulation.  I feel some further analysis is necessary here.  Additional comments are provided in the specific comments below.

**Section 3.1.3** still requires improvement.  I feel additional statistical analysis is necessary and I feel the authors' arguments at the end of the section need to be revised.  (These points are also relevant to **Section 3.2.2**.) In particular:
  (1) There is no assessment of the goodness of fit for the regression of $\delta^{18}O$ against temperature for the analysis shown in Figures 7 (c) and (d).  I feel this information is perhaps more important to the authors' arguments about the degree to which $\delta^{18}O$ and temperature are linked than the slope.  An analysis of the coefficient of determination ($R^2$ value) for $\delta^{18}O$ vs. temperature would be more appropriate in supporting the authors' points in this section.
  (2) The lower slope values for the temporal analysis shown in Figure 7 (d) are consistent with the lower coastal values shown in Figure 7(c), in that the seasonal analysis includes a larger range of variability in the high elevation areas.  Thus, a higher degree of temporal variability produces a lower slope.
  (3) I think the interpretation of the data here needs to be revised.  If the authors look more carefully at the correlation between $d^{18}O$ and temperature, they may find that temperature explains most of the variability in $\delta^{18}O$, while other processes explain additional variability.  The high $\delta^{18}O$ vs. temperature slope at higher elevations may be indicative that in these locations, temperature variability plays a lesser role than other factors, but here the degree of  $\delta^{18}O$ variability is small.  Therefore, overall, temperature still may

be the dominant factor in $\delta^{18}O$ variability. The authors should investigate this more carefully through a correlation analysis.

In **Section 3.2.1**, I am a bit concerned about the comparison shown in Figure 8(b) as the anomalies for the mid-Holocene relative to the pre-industrial are smaller than the difference between model and observations and the range of sub-grid-scale variability. I suppose the degree of temporal variability may be different from the degree of spatial variability. I would suggest performing a statistical test (e.g. t-test) to determine whether the anomaly is statistically different from 0 in both the case of observation and model results. This could be performed considering the interannual variability in both the Mid-Holocene and Pre-Industrial for both the model and observations.

**Specific Comments**

1. **Lines 10-11:** Suggest revising to "For this purpose, isotope-enabled simulations with the ECHAM5-wiso General Circulation Model (GCM) under present-day and the MPI-ESM-wiso GCM under mid-Holocene…" for clarity.
2. **Line 14:** Define GNIP.
3. **Lines 16-19:** Note here the spatial resolution of the GCM simulation (roughly). Again, provide some numbers to quantify the improvement in the agreement, and to support the statement.
4. **Lines 17-18:** This is a bit misleading. It should be noted that the 7 km simulation does not yield a substantial improvement overall, except in one area with complex terrain.
5. **Lines 23-25:** The statement "The correct $\delta^{18}O$ ratios are consequently already included but not resolved in the GCM simulations results…" is misleading. The correct values are not "included" in the GCM simulation. Rather the conclusion is that the discrepancies between the point measurements and GCM values are likely due sub-grid-scale variability not captured by the GCM. Suggest revising this to read: " $\delta^{18}O$ ratios are consequently not resolved in the GCM simulations …..
6. **Lines 97-99:** I think the authors should provide some documentation of the effect of this albedo change. Perhaps they can briefly document the improvements (through a set of tables for example) in a supplementary section or appendix.
7. **Line 139:** Can the authors briefly comment in the text on the choice of model domains? Why run the large domain RCM simulation over the Arctic and not simply run the RCM at high resolution over Greenland?
8. **Lines 166-168:** It is still unclear what the percentage of missing values is between 1940 and 2014 at each station and how this might affect the results. It would be helpful if the authors could estimate the uncertainty in the average value, which would place the model simulation in context.
9. **Line 174:** When the authors say the "observed isotope ratios are compared with simulated yearly mean $\delta^{18}O$ values in precipitation", it sounds as if the

annual modeled values are being compared to observed annual values, whereas only interannual averages are compared.  Suggest changing "the observed isotope ratios…" to read "we compute modeled annual mean $\delta^{18}O$ values and compared the multi-year 2008-2014 model mean to the observed values."

10. **Line 175:** Change "calculation of this yearly mean" to "calculation of the yearly modeled mean" for clarity.

11. **Table 2 caption:** Suggest revising to "…blue numbers in parentheses indicate mid-Holocene values." for clarity.

12. **Figure 2 caption:**  Note that the solid black line is the 1:1 line in (a) and (b).

13. **Lines 214-219:**  It is not clear how much of an improvement the RCM provides.  I suggest providing further details, for example what is the bias for the RCM and what is the bias for the GCM? What is the bias at the stations with poor agreement with the GCM and what is the bias with the RCM, and the same for the stations with a good agreement for the GCM initially?   Can the authors evaluate whether the change in the bias is statistically significant?  Also, the authors could compute the root mean squared error for the RCM and GCM simulations.  From this information the reader can better understand the degree of improvement associated with the RCM.

14. **Line 233:** If the authors can provide a bit more evidence that there is a significant improvement in agreement with observations when employing the RCM (as noted above), the authors could reiterate here at the end of the paragraph that despite the increased bias in northern Greenland, there is an overall improvement associated with the RCM simulation.

15. **Lines 283-285:**  This may be true at locations 11-13, where the bias is larger in COSMO_iso than in ECHAM5-wiso, but at locations 9 and 10, COSMO_iso is not very different from ECHAM5-wiso.  Suggest revising to make clear that the southward shift explains part but not all of the differences; e.g. "this is likely partially associated with the southward shift…"

16. **Line 287:**  Suggest changing "not covered within" to "fall outside of the range of".

17. **Line 288:** "increase the accuracy" is unclear.  Suggest changing to read "further downscaling… does not substantially change the simulated isotopic ratio spread…"

18. **Line 313:** Remove "the increase in" before "the spatial isotopic ratio variability".

19. **Line 317:** Suggest changing to "In central Greenland, surface temperature variability is very low (Figure 7b)."

20. **Line 321:** There is not increased $\delta^{18}O$ variability in central Greenland.  It is lower than along the coast but is relatively high compared with the temperature variability.  Please revise.

21. **Line 328-330:**  The lower slope does not necessarily imply a poor correlation.  Despite the higher slope in the spatial analysis, the correlation could be lower in these locations, while the correlation might actually be higher in the case of a lower slope, given the larger range of variability.  I

suggest computing the $R^2$ value for the linear regression for all grid cells as this will provide an indication of the degree of correlation. It is not clear whether the final statement that interannual temperature variations have a small impact on $\delta^{18}O$ variability is correct. To the contrary, temperature may be found to be the dominant factor in $\delta^{18}O$ variability both spatially and temporally if a more complete analysis is conducted.

22. **Lines 339-340:** Suggest revising to read "simulated mid-Holocene $\delta^{18}O$ ratios with comparison to observed mid-Holocene $\delta^{18}O$ values."

23. **Lines 348-349:** Suggest changing to: "For COSMO_iso_50km, the deviation of $d^{18}O$ values relative to observations are opposite in sign compared with MPI-ESM-wiso at all locations except Renland."

24. **Lines 352-354:** Suggest revising to "However, when the spatial isotopic ratio variability within MPI-ESM-wiso grid cells simulated by COSMO_iso_50km isotopic ratios is taken into account, the model results are in agreement with the isotopic ratios of the ice core samples."

25. **Line 371:** Please quantify "very small".

26. **Line 377:** Add "for the mid-Holocene" after "COSMO_iso_50km simulation" for clarity.

27. **Line 389-390:** Again, this conclusion is problematic because the authors are examining the slope, but not considering the correlation between $\delta^{18}O$ and temperature.

28. **Line 399:** These results are interesting, but I'm not sure they are so remarkable, given that temperature is expected to vary with elevation, and $\delta^{18}O$ seems to follow a similar pattern, being somewhat temperature dependent. They do point to a strong local influence on the spatial variability in $\delta^{18}O$.

29. **Line 403:** This seems a bit exaggerated. Clearly there is some difference with respect to the present-day simulation, and therefore the results are not entirely independent of the boundary conditions. I would suggest revising to read "...not strongly dependent on the oceanic boundary conditions."

30. **Line 415:** Suggest changing "already leads to" to "produces".

31. **Line 417:** Again define "considerably reduced" by providing some quantification. This is also not always the case as the results show.

32. **Lines 418-419:** I think the authors should note here the lack of improvement when increasing to 7 km.

33. **Line 424:** Not sure what is meant by "as it was simulated by". Possibly change to "as was the case in a similar study by Sjolte et al. (2011)"?

34. **Lines 452-456:** I agree with this statement, but it seems to contradict the authors' previous statements that there is not a strong relationship between temporal variations in temperature and $\delta^{18}O$, which was suggested based on the low $\delta^{18}O$-temperature slope. As discussed above, the authors should examine the correlation between temperature and $\delta^{18}O$ in order to determine the strength of that relationship, as well as to confirm the spatial relationships discussed here.

35. **Line 458-459:** Again, this contradicts the previous statement.

**Technical Corrections**

1. **Line 46:** Change "warming, in more detail" to "warming in more detail".
2. **Line 60:** Suggest changing to read "not able to quantitatively reproduce regional changes in isotope ratios"
3. **Line 63:** Change "ratios in precipitation,  by a regional" to "ratios in precipitation through a regional".
4. **Line 66:** Change "presented study" to "present study".
5. **Line 81:** Change "Holocene conditions, is performed" to "Holocene conditions is performed".
6. **Line 119:** Remove "have been suggested" before "(Gurney and Lawrence, 2004)".
7. **Line 121:** Change "just a simplified" to "a simplified".
8. **Line 173:** Change "Since both, snow pit" to "Since both snow pit…"
9. **Line 191:** Change "parameter" to "parameters".
10. **Line 196:** Change "Both, simulated" to "Both simulated"
11. **Line 210:** Change "is able to reflect" to "is able to reproduce".
12. **Lines 269-270:**  This sentence could be worded more clearly.  Suggest revising to: "Despite the lack of improvement in the point to grid-cell comparison, higher resolved RCM simulations allow the subgrid-scale variability of $\delta^{18}O$ within GCM grid boxes to be simulated and compared to observed $\delta^{18}O$ values."
13. **Lines 271-272:** Suggest revising to read: "Thus, in the following sections, snow pit samples are no longer solely compared…"
14. **Line 300:** Change "how strong" to "how strongly"
15. **Line 324:** Remove comma after "air mass".  Change "increase there the isotopic variability" to "increase the isotopic variability there."
16. **Line 328:** Change "in accordance" to "in agreement".
17. **Line 330:** Change "lowly correlated" to "poorly correlated".
18. **Line 334:** Suggest revising "COSMO_iso_50km is not anymore…" to "COSMO_iso_50km is driven by MPI-ESM-wiso rather than COSMO_iso_50km."
19. **Line 341:** Change "differences of" to "differences between".
20. **Line 342:** Change "to the observed" to "and the observed".
21. **Line 345:** Change "deviates only about 1 ‰ to the observations" to "deviates only by about 1 ‰ relative to the observation".
22. **Line 358:** Change to " the observed ratios derived from ice cores are subtracted from the simulated $\delta^{18}O$ ratios."
23. **Line 361:** Change "differences to" to "differences with respect to".  Change "anomalies of the MPI-ESM-wiso simulation to the pre-industrial" to "anomalies of the MPI-ESM-wiso simulation relative to pre-industrial"

24. **Line 362-363:** Change to "shown in red dots" to "shown as red points." For clarity, change "the observed mid-Holocene-PI" to "the observed anomalies for the mid-Holocene relative to present-day are shown as orange points."
25. **Figure 8 (b):** I would suggest changing the title to "Mid-Holocene anomalies (relative to PI)", and changing the caption to "MPI-ESM-wiso" and "observed". I would also suggest changing one set of points to be a different style to make the figure more easily readable.
26. **Line 368:** Add a comma after "especially during the summer".
27. **Line 370:** Add "and" after "slightly underestimated,"
28. **Line 377:** Change "for whole Greenland" to "for all of Greenland".
29. **Line 381:** Add "and" before "the GRIP and GISP2".
30. **Line 406:** I think this should read "The locations of the ice core samples are shown in green."
31. **Line 410:** Change "deviations to" to "deviations from".
32. **Line 425:** Add a comma after "rather".
33. **Line 426:** Remove "But" before "all in all"
34. **Line 431:** Change "with even an" to "and even an"
35. **Line 435:** Remove "Now," before "by analysing".
36. **Line 438:** Remove comma after "applies for both".
37. **Line 442:** Change "to reproduce" to "in reproducing".
38. **Lines 448-449:** Change "spatial variability pattern of" to "patterns of spatial variability in".
39. **Line 450:** Change "variability patterns" to "patterns of variability".
40. **Line 457:** Change "structures" to "patterns"?
41. **Line 466:** Change "and their deviations to" to "and understanding their deviations from".
42. **Line 467:** Remove comma after "regions".

---

## Referee Report (RR2)

**Review:  The dependency of the $\delta^{18}O$ discrepancy between ice cores and model simulations on the spatial model resolution**

Marcus Breil, E. Christner, M. Karremann, A. Cauquoin, M. Werner, G. Schädler

I have re-read the manuscript and the authors' responses and I find that the authors have adequately responded to the reviewers' concerns.  I feel the manuscript could benefit from some editing to reduce wordiness.  Technical corrections are suggested below.

**Technical Corrections**
1. **Line 10:** Change "with ECHAM5-wiso…" to "with the ECHAM5-wiso"
2. **Line 11:** Change "and MPI-ESM-wiso…" to "and the MPI-ESM-wiso"
3. **Line 22:** Change "improvements in" to "improvement in"
4. **Line 27:** Remove comma after "processes"
5. **Line 208:** Change "climatological parameter" to "climatological parameters"
6. **Line 209:** Change "parameter" to "parameters".
7. **Line 242:** Change "arctic" to "Arctic".
8. **Line 245:** Change "systematic deviations…are simulated." to "are systematic deviations to the observations simulated."
9. **Line 252:** Change "But in all," to "But in general,"
10. **Lines 260-261:** The "simulation: 2011, observation: mean present values" is confusing. Is this true for all the simulations shown in this figure? Please clarify.
11. **Line 264:** Change "in comparison to" to "with respect to"
12. **Line 272:** Remove "also" after "modeling,"
13. **Line 276:** Change "and the corresponding" to "versus the corresponding".
14. **Line 293:** Change "samples location" to "sample locations".
15. **Lines 299-300:** Change "In this spatial isotopic ratio variability," to "This spatial isotopic ratio variability includes".
16. **Line 301:** Remove "are included".
17. **Line 326:** Change "explicitly analyzed" to "explicitly shown"
18. **Lines 361-362:** I think "rather than COSMO_iso_50km" should be changed to "rather than ECHAM5-wiso".
19. **Line 370:** Change "four Greenland" to "the four Greenland".
20. **Lines 372-373:** Change "at Renland" to "and at Renland"
21. **Line 374:** Change "for NGRIP…" to "and for NGRIP.."
22. **Line 379:** Change "median of the isotope ratio distribution" to "median of the isotope ratio difference distribution".
23. **Line 383:** Change "blue points, the" to "blue points, and the"
24. **Line 405:** Change "Mid-Holocenen" to "Mid-Holocene".
25. **Line 418:** Change "in principle" to "in general".
26. **Line 431:** Change "dependent of" to "dependent on".
27. **Line 475:** Change "temperatures reconstructions" to "temperature reconstructions".
28. **Line 477:** Change "these pattern" to "these patterns".
29. **Line 484:** Change "partially be" to "be partially"

**30. Lines 487-488:** Suggest changing to read: "But in comparison to the spatial $\delta^{18}O$-temperature slope, the interannual $\delta^{18}O$-temperature slope is rather small."

**31. Table 2:** Sometime the decimal is used in the numbers, other times the comma (e.g. 26,73 vs. -27.38) Please make these consistent with the journal requirements.

**32. Figure S1 caption:** Note the period for the spatial comparison and make clear that "yearly mean values" refers to the temperature correlation.

---

## Author Response (AR2)

**Review**: The dependency of the d18O discrepancy between ice cores and model simulations on the spatial model resolution

Marcus Breil, Emanuel Christner, Alexandre Cauquoin, Martin Werner, Gerd Schädler

**General Comments**

I feel the authors have addressed some of the concerns from both reviewers, but I still have some concerns about the manuscript. In particular, although some additional quantitative evidence is now provided, I feel the authors have not supported their claims sufficiently with quantitative evidence, and additional quantitative analysis should be performed before the manuscript is published. If the authors find that the analysis does not support their claims, they should modify their conclusions to reflect this analysis. I think these changes will mostly be minor but since they have the potential to change the conclusions of the study I have selected major revisions.

- Dear Reviewer. Thank you very much for your constructive and very helpful comments. We think that you addressed some important issues. We hope that we are able to respond satisfactorily to your comments and clear the open issues you raised.

Some remaining general comments are:

In Section 3.1.2 the authors have computed the average reduction in bias for the RCM simulation. But it is still unclear how much of an improvement this represents. Is this a statistically significant difference? Has the root mean squared error also been reduced in the RCM simulation. I feel some further analysis is necessary here. Additional comments are provided in the specific comments below.

- the root mean squared error for COSMO_50km is reduced by 0.98 ‰, for COSMO_7km by 0.83 ‰. These improvements are statistically significant at the 95% level. This is now mentioned in the text (line 232-238):

"In COSMO_iso_50km the root mean squared error (RMSE) is reduced by 0.98 ‰ over all snow pit samples (the RMSE of ECHAM5-wiso is 2.42 ‰ and the RMSE of COSMO_iso_50km is 1.44 ‰). This RMSE reduction is significant at the 95 % level, assessed by performing a t-test. Especially for the snow pit samples for which ECHAM5-wiso exhibits strong deviations from the observed $\delta^{18}O$ values (1,3,4,6,7,8,9,10,16,17,19; see Table 2), a regional downscaling with COSMO_iso_50km reduces the bias considerably (Figure 4). For these stations, the RMSE of ECHAM5-wiso of 3.09 ‰ is reduced by 1.65 ‰ to 1.44 ‰. But for snow pit samples at which ECHAM5-wiso has already a high agreement with the observations (2,5,11,13,14), COSMO_iso_50km increases the RMSE from 0.34 ‰ to 1.51 ‰."

Section 3.1.3 still requires improvement. I feel additional statistical analysis is necessary and I feel the authors' arguments at the end of the section need to be revised. (These points are also relevant to Section 3.2.2.) In particular:

(1)There is no assessment of the goodness of fit for the regression of d18O against temperature for the analysis shown in Figures 7 (c) and (d). I feel this information is perhaps more important to the authors' arguments about the degree to which d18O and temperature are linked than the slope. An analysis of the coefficient of determination (R2value) for d18O vs. temperature would be more appropriate in supporting the authors' points in this section.

- we calculated the temporal (left) as well as the spatial (right) correlations (R² value) for d18O vs. temperature. The results are shown in the figure below and the figure is additionally included in Supplementary Material (S1).

[Figure]

Figure: temporal (left) and spatial (right) correlations (R² value) for d18O vs. temperature for the COSMO_iso_50km present-day simulation.

(2)The lower slope values for the temporal analysis shown in Figure 7 (d) are consistent with the lower coastal values shown in Figure 7(c), in that the seasonal analysis includes a larger range of variability in the high elevation areas. Thus, a higher degree of temporal variability produces a lower slope.

- you are right, a comparatively low (or better moderate) temporal d18O-temperature slope could also be associated with a large variability for both, d18O and temperature and thus a high correlation, as seen for the spatial slopes in coastal areas (figure 7c and figure above (right)). But the temporal slope shown in figure 7d is very low, while the spatial slope is moderate at the coastline (this is now explicitly mentioned in the manuscript). This fact already indicates that the temperature variability is larger than the d18O variability and that the correlation is reduced. This is also demonstrated by the analysis of the R² values in the figure above (left). But with respect to your detailed explanations, the text is adapted as follows (line 352-357):

"That means that the interannual $\delta^{18}O$ variability is less pronounced than the interannual surface temperature variability and thus, also the correlation between both quantities is not as strong as for the spatial $\delta^{18}O$ variability and the spatial temperature variability (Figure S1b in Supplementary Material). The mean spatial correlation over Greenland is 0.61 compared to 0.25 for the mean temporal correlation. The impact of interannual surface temperature variations on the temporal $\delta^{18}O$ variability in Greenland is therefore not as dominant as for the spatial $\delta^{18}O$ variability."

(3)I think the interpretation of the data here needs to be revised. If the authors look more carefully at the correlation between d18O and temperature, they may find that temperature explains most of the variability in d18O, while other processes explain additional variability. The high d18O vs. temperature slope at higher elevations may be indicative that in these locations, temperature variability plays a lesser role than other factors, but here the degree of d18O variability is small. Therefore, overall, temperature still may be the dominant factor in d18O variability. The authors should investigate this more carefully through a correlation analysis.

- in general, we agree. Temperature could still be the dominant factor for d18O variability. This is partly shown in the analysis of the spatial R² values (figure above (right)). The correlation is highest at the coast and less pronounced in Central Greenland, but still on a moderate level. Therefore, the respective statement in the manuscript is adapted as follows (line 341-348):

"The moderate spatial $\delta^{18}O$-temperature slope at the coastline (Figure 7c) is therefore a result of a high surface temperature variability in this region counteracting the high $\delta^{18}O$ variability in the slope calculation. There, the correlation between both quantities is consequently high (Figure S1a in

Supplementary Material, mainly between 0.7 and 0.99). In Central Greenland, the spatial $\delta^{18}$O-temperature slope is further increased due to the relatively high $\delta^{18}$O variability compared with the surface temperature variability. Therefore, the spatial distribution of $\delta^{18}$O cannot be solely explained by land surface processes and the associated spatial temperature variability. The additional spatial $\delta^{18}$O variability consequently must be caused by dynamic atmospheric processes. In this way, isotopic ratios based on atmospheric fractionation processes along the trajectory of an air mass are transported to Central Greenland and increase the isotopic variability there."

In Section 3.2.1, I am a bit concerned about the comparison shown in Figure 8(b)as the anomalies for the mid-Holocene relative to the pre-industrial are smaller than the difference between model and observations and the range of sub-grid-scale variability. I suppose the degree of temporal variability may be different from the degree of spatial variability. I would suggest performing a statistical test (e.g. t-test) to determine whether the anomaly is statistically different from 0 in both the case of observation and model results. This could be performed considering the interannual variability in both the Mid-Holocene and Pre-Industrial for both the model and observations.
- according to your suggestions, we performed a t-test on the anomalies of the Mid-Holocene conditions relative to the pre-industrial conditions. The results of this statistical test showed that the anomalies are not significant at the 95 % level. This is now mentioned in the text (line 397-399):

"But overall, the biases of the MPI-ESM-wiso mid-Holocene-PI model anomalies to the observed mid-Holocene-PI anomalies are for all ice cores small and statistically not significant at the 95 % level (assessed by performing a t-test)."

**Specific Comments**

1.Lines 10-11: Suggest revising to "For this purpose, isotope-enabled simulations with the ECHAM5-wiso General Circulation Model (GCM) under present-day and the MPI-ESM-wiso GCM under mid-Holocene..." for clarity.
- is changed according to your suggestion.

2.Line 14: Define GNIP.
- is now defined

3.Lines 16-19: Note here the spatial resolution of the GCM simulation (roughly). Again, provide some numbers to quantify the improvement in the agreement, and to support the statement.
- is now noted.

4.Lines 17-18: This is a bit misleading. It should be noted that the 7 km simulation does not yield a substantial improvement overall, except in one area with complex terrain.
- is changed according to your suggestion.

5.Lines 23-25: The statement "The correct d18O ratios are consequently already included but not resolved in the GCM simulations results..." is misleading. The correct values are not "included" in the GCM simulation. Rather the conclusion is that the discrepancies between the point measurements and GCM values are likely due sub-grid-scale variability not captured by the GCM. Suggest revising this to read: "d18O ratios are consequently not resolved in the GCM simulations .....
- is changed according to your suggestion.

6.Lines 97-99: I think the authors should provide some documentation of the effect of this albedo change. Perhaps they can briefly document the improvements (through a set of tables for example) in a supplementary section or appendix.

- a detailed documentation about the impact of the snow albedo change is provided by Karremann & Schädler (2021). This is now mentioned in the manuscript.

7.Line 139:Can the authors briefly comment in the text on the choice of model domains? Why run the large domain RCM simulation over the Arctic and not simply run the RCM at high resolution over Greenland?
- in order to guarantee at the boundaries of the regional model domain a physically consistent transition between the coarse model resolution of the GCM and the fine model resolution of the RCM, the model resolution is step-wise increased. This procedure is called nesting. This is now mentioned in the manuscript (line 131-137):

"In order to guarantee at the boundaries of the regional model domain a physically consistent transition between the coarse model resolution of the GCM and the fine model resolution of the RCM, the model resolution is step-wise increased. This procedure is called nesting. In a first nesting step, the spatial resolution of COSMO_iso is set to 0.44° x 0.44°, corresponding to 50 km x 50 km in rotated coordinates (COSMO_iso_50km). In a second nesting step, an additional COSMO_iso simulation with a spatial resolution of 0.0625° x 0.0625° (corresponding to about 7 km × 7 km) for Greenland (COSMO_iso_7km) is nested in the COSMO_iso_50km simulation."

8.Lines 166-168: It is still unclear what the percentage of missing values is between 1940 and 2014 at each station and how this might affect the results. It would be helpful if the authors could estimate the uncertainty in the average value, which would place the model simulation in context.
- The standard deviations of the observed yearly d18O values at the snow pit samples is now included in table 2.

9.Line 174: When the authors say the "observed isotope ratios are compared with simulated yearly mean d18O values in precipitation", it sounds as if the annual modeled values are being compared to observed annual values, whereas only interannual averages are compared. Suggest changing "the observed isotope ratios..." to read "we compute modeled annual mean d18O values and compared the multi-year 2008-2014 model mean to the observed values."
- is changed according to your suggestion.

10.Line 175:Change "calculation of this yearly mean" to "calculation of the yearly modeled mean" for clarity.
- is changed according to your suggestion.

11.Table 2 caption: Suggest revising to "...blue numbers in parentheses indicate mid-Holocene values." for clarity.
- is changed according to your suggestion.

12.Figure 2caption: Note that the solid black line is the 1:1 line in (a) and (b).
- is changed according to your suggestion.

13.Lines 214-219: It is not clear how much of an improvement the RCM provides. I suggest providing further details, for example what is the bias for the RCM and what is the bias for the GCM? What is the bias at the stations with poor agreement with the GCM and what is the bias with the RCM, and the same for the stations with a good agreement for the GCM initially? Can the authors evaluate whether the change in the bias is statistically significant? Also, the authors could compute the root mean squared error for the RCM and GCM simulations. From this information the reader can better understand the degree of improvement associated with the RCM.
- we agree, we should have provided further statistical details. This is now done in the revised manuscript (line 232-238):

"In COSMO_iso_50km the root mean squared error (RMSE) is reduced by 0.98 ‰ over all snow pit samples (the RMSE of ECHAM5-wiso is 2.42 ‰ and the RMSE of COSMO_iso_50km is 1.44 ‰). This RMSE reduction is significant at the 95 % level, assessed by performing a t-test. Especially for the snow pit samples for which ECHAM5-wiso exhibits strong deviations from the observed $\delta^{18}O$ values (1,3,4,6,7,8,9,10,16,17,19; see Table 2), a regional downscaling with COSMO_iso_50km reduces the bias considerably (Figure 4). For these stations, the RMSE of ECHAM5-wiso of 3.09 ‰ is reduced by 1.65 ‰ to 1.44 ‰. But for snow pit samples at which ECHAM5-wiso has already a high agreement with the observations (2,5,11,13,14), COSMO_iso_50km increases the RMSE from 0.34 ‰ to 1.51 ‰."

14. Line 233: If the authors can provide a bit more evidence that there is a significant improvement in agreement with observations when employing the RCM (as noted above), the authors could reiterate here at the end of the paragraph that despite the increased bias in northern Greenland, there is an overall improvement associated with the RCM simulation.
- the improvements of the regional climate simulations are statistically significant (see comment above and the response to the major comment regarding Section 3.1.2). In the revised manuscript we mention now the overall improvements of the RCM simulation (line 252-253):

"But in all, COSMO_iso_50km yields an overall improvement in simulating the yearly mean $\delta^{18}O$ values compared to ECHAM5-wiso"

15. Lines 283-285: This may be true at locations 11-13, where the bias is larger in COSMO_iso than in ECHAM5-wiso, but at locations 9 and 10, COSMO_iso is not very different from ECHAM5-wiso. Suggest revising to make clear that the southward shift explains part but not all of the differences; e.g. "this is likely partially associated with the southward shift..."
- is changed according to your suggestion.

16. Line 287: Suggest changing "not covered within" to "fall outside of the range of".
- is changed according to your suggestion.

17. Line 288: "increase the accuracy" is unclear. Suggest changing to read "further downscaling... does not substantially change the simulated isotopic ratio spread..."
- is changed according to your suggestion.

18. Line313: Remove "the increase in" before "the spatial isotopic ratio variability".
- is changed according to your suggestion.

19. Line 317: Suggest changing to "In central Greenland, surface temperature variability is very low (Figure 7b)."
- is changed according to your suggestion.

20. Line 321: There is not increased d18O variability in central Greenland. It is lower than along the coast but is relatively high compared with the temperature variability. Please revise.
- is revised as follows (line 344-345):

"In Central Greenland, the spatial $\delta^{18}O$-temperature slope is further increased due to the relatively high $\delta^{18}O$ variability compared with the surface temperature variability."

21. Line 328-330: The lower slope does not necessarily imply a poor correlation. Despite the higher slope in the spatial analysis, the correlation could be lower in these locations, while the correlation might actually be higher in the case of a lower slope, given the larger range of variability. I suggest

computing the R2 value for the linear regression for all grid cells as this will provide an indication of the degree of correlation. It is not clear whether the final statement that interannual temperature variations have a small impact on d18O variability is correct. To the contrary, temperature may be found to be the dominant factor in d18O variability both spatially and temporally if a more complete analysis is conducted.

- again we agree with you that the interrelation between d18O and temperature variability is not necessarily as described in the manuscript (see response to major comments regarding section 3.1.3). But for the temporal correlations only low R² values between d18O and temperature are calculated. But with respect to your detailed explanations, we adapted the statement as follows (line 352-357):

"That means that the interannual $\delta^{18}$O variability is less pronounced than the interannual surface temperature variability and thus, also the correlation between both quantities is not as strong as for the spatial $\delta^{18}$O variability and the spatial temperature variability. The impact of interannual surface temperature variations on the temporal $\delta^{18}$O variability in Greenland is therefore not as dominant as for the spatial $\delta^{18}$O variability."

22.Lines 339-340: Suggest revising to read "simulated mid-Holocene d18O ratios with comparison to observed mid-Holocene d18O values."
- is changed according to your suggestion.

23.Lines 348-349: Suggest changing to: "For COSMO_iso_50km, the deviation of d18O values relative to observations are opposite in sign compared with MPI-ESM-wiso at all locations except Renland."
- is changed according to your suggestion.

24.Lines 352-354: Suggest revising to "However, when the spatial isotopic ratio variability within MPI-ESM-wiso grid cells simulated by COSMO_iso_50km isotopic ratios is taken into account, the model results are in agreement with the isotopic ratios of the ice core samples."
- is changed according to your suggestion.

25.Line 371: Please quantify "very small".
- this statement is removed from the manuscript and replaced by a statement of the statistical significance (line 397-399). Additionally, we added at several places in the text concrete numbers to substantiate and quantify statements like this (see lines 324, 343-344, 355, 415-417).

26.Line 377: Add "for the mid-Holocene" after "COSMO_iso_50km simulation" for clarity.
- is added.

27.Line 389-390: Again, this conclusion is problematic because the authors are examining the slope, but not considering the correlation between d18O and temperature.
- see the correlation analysis performed for the response to the major comments. The text is adapted as follows (line 418-419):

"But in principle, the influence of interannual surface temperature variations on the temporal $\delta^{18}$O variability in the mid-Holocene is not as dominant as for the spatial variability."

28.Line 399: These results are interesting, but I'm not sure they are so remarkable, given that temperature is expected to vary with elevation, and d18O seems to follow a similar pattern, being somewhat temperature dependent. They do point to a strong local influence on the spatial variability in d18O.

- the statement is removed in the revised manuscript.

29.Line 403: This seems a bit exaggerated. Clearly there is some difference with respect to the present-day simulation, and therefore the results are not entirely independent of the boundary conditions. I would suggest revising to read "...not strongly dependent on the oceanic boundary conditions."
- is changed according to your suggestion.

30.Line 415: Suggest changing "already leads to" to "produces".
- is changed according to your suggestion.

31.Line 417: Again define "considerably reduced" by providing some quantification. This is also not always the case as the results show.
- the text is adapted as follows (line 443-446):

"Especially in regions where the global ECHAM5-wiso model, which has been used to derive necessary forcing fields for the COSMO_iso simulations, deviates strongly from the observed $\delta^{18}O$ values, the RMSE is significantly reduced by 1.65 ‰ for regional climate simulation with COSMO_iso "

32.Lines 418-419: I think the authors should note here the lack of improvement when increasing to 7 km.
- the lack of improvement in highly resolved simulations is now mentioned in the manuscript (line 449-450):

"But for the rest of Greenland, highly resolved regional climate simulations do not yield further improvements"

33.Line 424: Not sure what is meant by "as it was simulated by". Possibly change to "as was the case in a similar study by Sjolte et al. (2011)"?
- is changed according to your suggestion.

34.Lines 452-456: I agree with this statement, but it seems to contradict the authors' previous statements that there is not a strong relationship between temporal variations in temperature and d18O, which was suggested based on the low d18O-temperature slope. As discussed above, the authors should examine the correlation between temperature and d18O in order to determine the strength of that relationship, as well as to confirm the spatial relationships discussed here.
- the original statement is qualified in the revised manuscript. Now it is only stated that the temporal interrelations between temperature and d18O are not as dominant as the spatial ones (line 352-357). See response to the major comments.

35.Line 458-459: Again, this contradicts the previous statement.
- see previous comment.

**Technical Corrections**

1.Line 46: Change "warming, in more detail" to "warming in more detail".
- is changed.

2.Line 60: Suggest changing to read "not able to quantitatively reproduce regional changes in isotope ratios"
- is changed.

3.Line 63: Change "ratios in precipitation, by a regional" to "ratios in precipitation through a regional".

- is changed.

4.Line 66: Change "presented study" to "present study".

- is changed.

5.Line 81: Change "Holocene conditions, is performed" to "Holocene conditions is performed".

- is changed.

6.Line 119: Remove "have been suggested" before "(Gurney and Lawrence, 2004)".

- is removed.

7.Line 121: Change "just a simplified" to "a simplified".

- is changed.

8.Line 173: Change "Since both, snow pit" to "Since both snow pit...".

- is changed.

9.Line 191: Change "parameter" to "parameters".

- is changed.

10.Line 196: Change "Both, simulated" to "Both simulated"

- is changed.

11.Line 210: Change "is able to reflect" to "is able to reproduce".

- is changed.

12.Lines 269-270: This sentence could be worded more clearly. Suggest revising to: "Despite the lack of improvement in the point to grid-cell comparison, higher resolved RCM simulations allow the subgrid-scale variability of d18O within GCM grid boxes to be simulated and compared to observed d18O values."

- is replaced.

13.Lines 271-272: Suggest revising to read: "Thus, in the following sections, snow pit samples are no longer solely compared..."

- is changed.

14.Line 300: Change "how strong" to "how strongly"

- is changed.

15.Line 324: Remove comma after "air mass". Change "increase there the isotopic variability" to "increase the isotopic variability there."

- is changed.

16.Line 328: Change "in accordance" to "in agreement".

- is changed.

17.Line 330: Change "lowly correlated" to "poorly correlated".

- is changed.

18.Line 334: Suggest revising "COSMO_iso_50km is not anymore..." to "COSMO_iso_50km is driven by MPI-ESM-wiso rather than COSMO_iso_50km."
- is revised.

19.Line 341: Change "differences of" to "differences between".
- is changed.

20.Line 342: Change "to the observed" to "and the observed".
- is changed.

21.Line 345:Change "deviates only about 1 ‰ to the observations" to "deviates only by about 1 ‰ relative to the observation".
- is changed.

22.Line 358: Change to "the observed ratios derived from ice cores are subtracted from the simulated d18O ratios."
- is changed.

23.Line 361:Change "differences to" to "differences with respect to". Change "anomalies of the MPI-ESM-wiso simulation to the pre-industrial" to "anomalies of the MPI-ESM-wiso simulation relative to pre-industrial"
- are changed.

24.Line 362-363: Change to "shown in red dots" to "shown as red points." For clarity, change "the observed mid-Holocene-PI" to "the observed anomalies for the mid-Holocene relative to present-day are shown as orange points."
- are changed

25.Figure 8 (b): I would suggest changing the title to "Mid-Holocene anomalies (relative to PI)", and changing the caption to "MPI-ESM-wiso"and "observed". I would also suggest changing one set of points to be a different style to make the figure more easily readable.
- the figure is changed according to your suggestions.

26.Line 368: Add a comma after "especially during the summer".
- is added.

27.Line 370: Add "and" after "slightly underestimated,"
- is added.

28.Line 377: Change "for whole Greenland" to "for all of Greenland".
- is changed.

29.Line 381: Add "and" before "the GRIP and GISP2".
- is added.

30.Line 406: I think this should read "The locations of the ice core samples are shown in green."
- is changed.

31.Line 410: Change "deviations to" to "deviations from".
- is changed.

32.Line 425: Add a comma after "rather".

- is added.

33.Line 426: Remove "But" before "all in all"
- is removed.

34.Line 431: Change "with even an" to "and even an"
- is changed.

35.Line 435: Remove "Now," before "by analysing".
- is removed.

36.Line 438: Remove comma after "applies for both".
- is removed.

37.Line 442: Change "to reproduce" to "in reproducing".
- is changed.

38.Lines 448-449: Change "spatial variability pattern of" to "patterns of spatial variability in".
- is changed.

39.Line 450: Change "variability patterns" to "patterns of variability".
- is changed.

40.Line 457: Change "structures" to "patterns"?
- is changed.

41.Line 466: Change "and their deviations to" to "and understanding their deviations from".
- is changed.

42.Line 467: Remove comma after "regions".
- is removed.